# Regulation of shoot meristem shape by photoperiodic signaling and phytohormones during floral induction of Arabidopsis

Atsuko Kinoshita[1,2†], Alice Vayssières[1†], René Richter[1,3], Qing Sang[1], Adrian Roggen[1], Annabel D van Driel[1], Richard S Smith[1‡], George Coupland[1]*

[1]Max Planck Institute for Plant Breeding Research, Cologne, Germany; [2]Department of Biological Sciences, Tokyo Metropolitan University, Hachioji, Japan; [3]School of Agriculture and Food, University of Melbourne, Melbourne, Australia

**Abstract** Floral transition, the onset of plant reproduction, involves changes in shape and identity of the shoot apical meristem (SAM). The change in shape, termed doming, occurs early during floral transition when it is induced by environmental cues such as changes in day-length, but how it is regulated at the cellular level is unknown. We defined the morphological and cellular features of the SAM during floral transition of *Arabidopsis thaliana*. Both cell number and size increased during doming, and these changes were partially controlled by the gene regulatory network (GRN) that triggers flowering. Furthermore, dynamic modulation of expression of gibberellin (GA) biosynthesis and catabolism enzymes at the SAM contributed to doming. Expression of these enzymes was regulated by two MADS-domain transcription factors implicated in flowering. We provide a temporal and spatial framework for integrating the flowering GRN with cellular changes at the SAM and highlight the role of local regulation of GA.

**\*For correspondence:**
coupland@mpipz.mpg.de

[†]These authors contributed equally to this work

**Present address:** [‡]John Innes Centre, Norwich Research Park, Norwich, United Kingdom

**Competing interests:** The authors declare that no competing interests exist.

## Introduction

In plants, all shoot tissues are derived from the shoot apical meristem (SAM), a group of cells at the apex of the plant that includes a population of self-renewing stem cells. Organ primordia are formed continuously on the flanks of the SAM, and these change in identity during growth and development. In the model plant *Arabidopsis thaliana*, leaf primordia are formed during vegetative growth, but at floral transition the developmental identity of the SAM changes to an inflorescence meristem, and it initiates the formation of floral primordia. This process of floral induction represents the first step in plant reproduction and is closely regulated by environmental cues and by the developmental stage of the plant. One of these environmental cues is day length or photoperiod, which in many species synchronizes reproduction with the changing seasons (*Andrés and Coupland, 2012*). For example, floral induction of *A. thaliana* occurs rapidly in response to exposure to long days (LDs). Here we define early cellular changes at the SAM during floral induction in response to LDs, and determine the contributions of genes that control flowering time or encode enzymes that regulate levels of the phytohormone gibberellin (GA).

The size and shape of the inflorescence meristem, and the regulation of cell division and cell size within it, have been studied in detail (*Gaillochet et al., 2017*; *Jones et al., 2017*; *Laufs et al., 1998*). However, the programmed alterations in shape of the SAM that occur during the transition from a vegetative to inflorescence meristem are less well understood.

During induction of flowering in many plant species, the SAM increases in size and takes on a domed shape prior to the production of flowers, and these changes are induced by exposure to

environmental cues (*Kwiatkowska, 2008*). The contribution of different regions of the meristem to doming has been analyzed. The SAM is considered to consist of three regions based on histological and functional analyses: the central zone (CZ), the peripheral zone (PZ), and the rib zone (RZ) (*Barton, 2010*; *Bowman and Eshed, 2000*). The CZ contains the stem cells, the PZ gives rise to primordia, and the RZ forms the stem. Cells in the CZ divide slowly, whereas cells in the PZ divide more frequently to produce organ primordia. During floral induction initiated by exposure to LDs, mitotic activity in the SAM of *A. thaliana* was measured by tritiated thymidine labeling and was found to increase in both the CZ and the PZ (*Jacqmard et al., 2003*). Similar conclusions were drawn from classical histological studies in *Sinapis alba* and *Helianthus annuus*, where increases in mitotic activity at the SAM upon floral induction reduced the difference in the rate of mitosis between the CZ and the PZ (*Bodson, 1975*; *Marc and Palmer, 1982*). Although these histological analyses identified the correlation between cell division frequency and enlargement of the SAM, the resulting cellular changes and the molecular mechanisms that underlie the phenomenon have not been determined. Recently, a study in *Solanum lycopersicum* demonstrated that increased expression of an anti-florigen, SELF PRUNING (SP), delayed floral transition and uncoupled doming of the SAM and floral development, whereas mutation of the florigen encoding gene *SINGLE-FLOWER TRUSS* (*SFT*) prevented doming at the morphological level (*Tal et al., 2017*). However, the contribution at the cellular level of florigen to doming of the SAM during floral transition remains to be determined.

Genetic analyses have defined a pathway that regulates flowering of *A. thaliana* in response to photoperiod (*Koornneef et al., 1991*; *Turck et al., 2008*). The paralogous proteins FLOWERING LOCUS T (FT) and TWIN SISTER OF FT (TSF) represent the output of this pathway (*Kardailsky et al., 1999*; *Kobayashi et al., 1999*; *Yamaguchi et al., 2005*). *FT* and *TSF* are transcribed specifically under LDs in the leaf vasculature, and the FT protein is transported to the SAM, where it interacts with 14-3-3 proteins and a bZIP transcription factor, FD, to promote the transcription of floral integrator genes (*Abe et al., 2005*; *Abe et al., 2019*; *Collani et al., 2019*; *Corbesier et al., 2007*; *Jaeger and Wigge, 2007*; *Mathieu et al., 2007*; *Romera-Branchat et al., 2020*; *Tamaki et al., 2007*; *Taoka et al., 2011*). The earliest known gene to respond directly to FT/FD at the SAM is *SUPPRESSOR OF OVEREXPRESSION OF CONSTANS 1* (*SOC1*) that encodes a MADS-domain type transcription factor (*Lee et al., 2000*; *Samach et al., 2000*). Subsequently, the FT/FD complex also activates the transcription of *APETALA1* (*AP1*), another gene encoding a MADS-domain transcription factor that confers floral identity on primordia on the flanks of the SAM (*Abe et al., 2005*; *Collani et al., 2019*; *Wigge et al., 2005*). On the other hand, flowering of *A. thaliana* is severely delayed under non-inductive short-day conditions, in which the MADS-domain transcription factor SHORT VEGETATIVE PHASE (SVP) inhibits flowering by reducing transcription of *FT*, *TSF*, and *SOC1* (*Hartmann et al., 2000*; *Jang et al., 2009*). The onset of flowering can be synchronized by transferring plants from SDs to LDs, and in the wild-type accession Col-0, the SAM of plants grown for 2 weeks under SDs becomes committed to floral induction around 5 days after transfer to LDs (*Torti et al., 2012*). The histological and transcriptional profiles of the SAM change dramatically as flowering is initiated and proceeds, but the mechanisms underlying how these histological changes occur and are initiated by environmental cues remain largely unknown (*Kwiatkowska, 2008*; *Laufs et al., 1998*; *Schmid et al., 2003*; *Torti et al., 2012*; *Uchida and Torii, 2019*).

GA promotes diverse biological processes, including cell elongation, cell division, and floral induction (*Yamaguchi, 2008*). The strongly GA-deficient *ga1-3* mutant of *A. thaliana* is late flowering under LDs and fails to flower under SDs (*Wilson et al., 1992*). Corresponding to this phenotype, the level of bioactive $GA_4$ increases strongly in shoot apices under SDs around the time of floral induction (*Eriksson et al., 2006*). Local GA biosynthesis is largely dependent on the activity of the 2-oxoglutarate-dependent dioxygenase (2ODD) enzymes GA20-oxidase (GA20ox) and GA3-oxidase (GA3ox) that convert the GA precursor $GA_{12}$ into bioactive $GA_4$ (*Yamaguchi, 2008*). Both are encoded by gene families in *A. thaliana*, and individual family members exhibit distinct expression patterns (*Han and Zhu, 2011*; *Mitchum et al., 2006*; *Plackett et al., 2012*). By contrast, the 2ODD enzyme GA2-oxidase (GA2ox) contributes to the inactivation of $GA_4$ or its precursors and regulates the concentration of bioactive GA in vivo (*Yamaguchi, 2008*). In *A. thaliana*, this enzyme is also encoded by a gene family, and individual genes are expressed in specific patterns (*Rieu et al., 2008a*). Bioactive GA is perceived by its receptor, GIBBERELLIN INSENSITIVE DWARF1 (GID1) (*Griffiths et al., 2006*; *Willige et al., 2007*). The GID1–GA complex interacts directly with DELLA proteins and induces their degradation by the 26S-proteasome pathway through the E3 ubiquitin

ligase SLEEPY (SLY) and SNEEZY (SNE) (*Ariizumi et al., 2011*; *Griffiths et al., 2006*; *McGinnis et al., 2003*; *Willige et al., 2007*). DELLA proteins are repressors of GA-mediated growth and development, and they interact with a broad range of transcription factors to alter their DNA binding properties or transcriptional activity (*Marín-de la Rosa et al., 2014*). In *A. thaliana* there are five DELLA proteins, of which REPRESSOR OF ga1-3 (RGA) and GA INSENSITIVE (GAI) are major repressors of growth and reproductive development (*Dill and Sun, 2001*; *Peng et al., 1997*; *Silverstone et al., 1998*). Transcription of GA biosynthesis and deactivating genes is regulated by DELLA proteins, demonstrating that these contribute to feedback and feedforward regulation of GA metabolism (*Hedden and Phillips, 2000*; *Zentella et al., 2007*).

GA levels are strongly repressed in the SAM of different plant species during vegetative development. Establishment and maintenance of the SAM involve KNOTTED1-LIKE homeobox (KNOX) transcription factors, such as KNOTTED1 (KN1) in maize and SHOOTMERISTEMLESS (STM) in *A. thaliana*, which are expressed in the SAM from the early stages of embryogenesis (*Bowman and Eshed, 2000*; *Long et al., 1996*; *Vollbrecht et al., 1991*). A major function of KNOX transcription factors is the reduction in GA levels in the SAM (*Bolduc and Hake, 2009*; *Hay et al., 2002*; *Jasinski et al., 2005*; *Sakamoto et al., 2001a*; *Sakamoto et al., 2001b*). In maize, KN1 directly binds to and activates expression of the *GA2ox1* gene that encodes a GA catabolic enzyme. Similarly, in Arabidopsis and tobacco, the activity of STM and its homologue NTH15, respectively, are associated with reduced expression of *GA20ox* genes that encode GA biosynthetic enzymes (*Hay et al., 2002*; *Jasinski et al., 2005*; *Sakamoto et al., 2001a*). Therefore, the concentration of GA is maintained at a low level in the SAM and this is proposed to be required for meristem activity. Indeed, the previous studies linking *KNOX* genes to GA levels in the SAM showed the specific expression of genes encoding GA2 oxidases and GA20 oxidases in the SAM and the leaf primordia, respectively (*Bolduc and Hake, 2009*; *Hay et al., 2002*; *Jasinski et al., 2005*; *Sakamoto et al., 2001a*; *Sakamoto et al., 2001b*). However, due to the resolution and sensitivity of the β-glucuronidase activity and in situ mRNA hybridization methods used to detect gene expression, it remains unclear in which domains of the SAM these genes are expressed and how dynamic their expression patterns are during development. Recently, several studies used tissue-specific promoters to express GA catabolic enzymes or DELLA proteins in the leaves and in the SAM separately, and these approaches suggested functions for GA in the SAM during flowering (*Bao et al., 2020*; *Galvão et al., 2012*; *Porri et al., 2012*). Furthermore, the mRNA of *GA20ox2* and GA levels were found to increase in apices of *A. thaliana* during floral transition (*Andrés et al., 2014*). Thus, the repression of GA mediated by KNOX proteins may be overcome during floral induction.

Here we analyze SAM shape and cellular content by confocal microscopy and demonstrate dynamic histological and cellular changes at the SAM during floral transition induced by LDs. Both the number and size of the cells in the SAM increased in the early stages of this process, and these changes are at least partially regulated by the photoperiodic flowering pathway and GA. Detailed observations of reporter lines revealed dynamic changes in expression patterns of genes encoding GA biosynthesis and catabolism enzymes in the SAM. Furthermore, the expression of these genes was found to be regulated by two MADS-domain transcription factors that play key roles in the control of flowering time. These results define roles for the photoperiodic flowering and phytohormone signaling pathways in the dynamic control of SAM properties during floral transition.

## Results

### Both cell division and elongation are enhanced in the SAM at floral transition

To understand better the histological changes at the SAM during photoperiod-induced flowering, we investigated the configuration of the cells on the surface of the SAM in plants grown under SDs for 2 weeks (2wSD), then either transferred to LDs or maintained in SDs for 7 days (3wSD) (*Figure 1A* and *Figure 1—figure supplement 1*). For this analysis, the SAM was defined as the region between the first developing primordia ($P_n$) and its boundaries were delimited by regions of negative Gaussian curvature (*Figure 1—figure supplement 2A*). In wild-type plants grown in noninductive SDs, the area of the SAM and the distance between primordia on opposite sides of the SAM gradually increased between 2wSD and 3wSD (*Figure 1B* and *Figure 1—figure supplement*

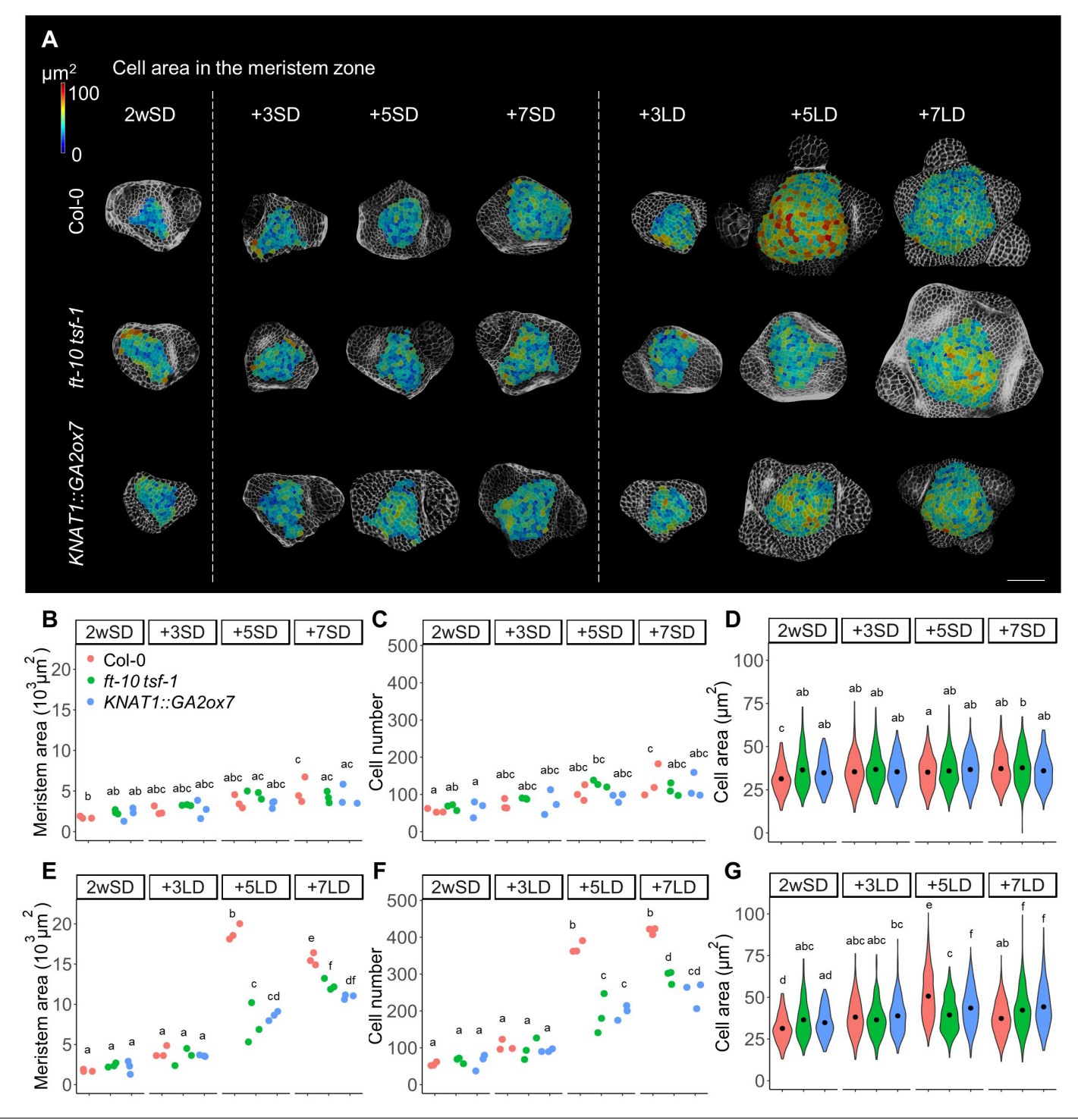

**Figure 1.** The number and size of the cells in the meristem increase during floral transition. (**A**) Heat-map quantification of cell area in the meristem region of Col-0, *ft-10 tsf-1*, and *KNAT1::GA2ox7* grown for 2 weeks in non-inductive short days (SDs) (2wSD) and observed for an additional 3SDs (+3SD), 5SDs (+5SD), or 7SDs (+7SD), or transferred to inductive long days (LDs) for 3LDs (+3LD), 5LDs (+5LD), or 7LDs (+7LD). Scale bars, 50 μm. (**B–G**) Quantification of the meristem area (**B and E**), the cell number (**C and F**), and the cell area in the meristem region (**D and G**) in Col-0, *ft-10 tsf-1*, and *KNAT1::GA2ox7* grown for 2 weeks in non-inductive SDs (2wSD) and observed for +3SD, +5SD, or +7SD (**B–D**) or transferred to +3LD, +5LD, or +7LD (**E–G**). Letters a–f in panels B–G show significant differences between conditions and genotypes (p<0.05, using ANOVA followed by Tukey's pairwise multiple comparisons), n = 3 apices.

The online version of this article includes the following source data and figure supplement(s) for figure 1:

*Figure 1 continued on next page*

*Figure 1 continued*

**Source data 1.** Original data of meristem area and cell number of each genotype for *Figure 1A–C,E, and F* and *Figure 1—figure supplement 1*.
**Source data 2.** Original data of cell size of each genotype for *Figure 1A,D, and G* and *Figure 1—figure supplement 2*.
**Figure supplement 1.** Doming of the shoot apical meristem (SAM) is observed specifically at floral transition.
**Figure supplement 2.** Larger increase in cell area in the PZ than in the CZ at floral transition.

*1A and B*). During this time, the number of cells in the SAM also slightly increased (*Figure 1C*). By contrast, when 2-week-old SD-grown plants were transferred to LDs, both SAM area and the number of cells in the SAM increased rapidly (*Figure 1E and F*). In particular, a dramatic increase in the number of cells in the SAM was observed between 3 and 5 days after transfer to LDs (*Figure 1F*). This increase in cell number was associated with doming of the SAM (*Figure 1—figure supplement 1A*). Later, between 5 and 7 days after transfer to LDs, the area of the SAM and the distance between primordia on opposite sides of the SAM decreased, although the number of cells in the SAM did not change significantly between these two time points (*Figure 1E and F* and *Figure 1—figure supplement 1C*). Therefore, we next examined the size of the cells in the SAM during floral transition. Statistical analysis showed that cell size in the SAM significantly increased between 3 and 5 days after transfer to LDs, and decreased again between 5 and 7 LDs (*Figure 1D and G*). To examine the difference in cell size in distinct regions of the SAM, we compared the size of cells in a central region (defined as zero to two cells from the central cell at the apex of the shoot) with that of cells in a ring surrounding this central region (three to five cells from the central cell) at each time point (*Figure 1—figure supplement 2A*). Cell size increased in response to LDs in both regions, but the effect was stronger in the outer ring area than in the central region (*Figure 1—figure supplement 2B*). These histological analyses suggest that as well as cell number, cell size increased in response to transfer from SDs to LDs and that the transient enlargement of the cells correlates with doming of the SAM during floral transition.

We next examined whether increases in the number and size of the cells in the SAM are associated with floral induction and activity of the photoperiodic flowering pathway. To this end, we analyzed the meristem of the *ft-10 tsf-1* mutant, in which flowering is not induced in response to LDs. Although the size of the SAM of *ft-10 tsf-1* was higher in plants transferred to LDs compared to those maintained in SDs, the number of cells in the SAM was significantly lower in *ft-10 tsf-1* mutants exposed to LDs than in wild-type Col-0 (*Figure 1A–C,E, and F*). Besides a small increase in cell size observed at seven LDs, no transient increase in cell size was observed in the *ft-10 tsf-1* mutant at five LDs (*Figure 1G*). Consistent with this observation, doming of the SAM was strongly delayed in this background, although the area of the SAM expanded laterally after transfer to LDs (*Figure 1—figure supplement 1A*).

Previous studies showed that GA regulates both cell division and elongation in the root apical meristem (*Achard et al., 2009*; *Ubeda-Tomás et al., 2008*). Therefore, we examined the role of GA in cell division and enlargement in the SAM during floral transition by analyzing the SAM of *KNAT1::GA2ox7* plants. This transgene reduces the level of GA precursor specifically in the SAM, but is not expressed directly in the epidermis (*Lincoln et al., 1994*; *Porri et al., 2012*; *Schomburg et al., 2003*). Both cell number and meristem size were reduced in *KNAT1::GA2ox7* after growth for 2wSD and transfer to 5 and 7 LDs, similar to *ft-10 tsf-1* (*Figure 1A–F*). However, the meristem of the *KNAT1::GA2ox7* line was morphologically different from that of the *ft-10 tsf-1* mutant, with a specific reduction in meristem width (*Figure 1—figure supplement 1A,D, and E*). In addition, cell area in the meristematic region of this line was larger than for the *ft-10 tsf-1* mutant after five LDs (*Figure 1G*). Furthermore, no difference was observed between the cell area in the inner central region or the surrounding ring region at five LDs in the *KNAT1::GA2o × 7* line or in *ft-10 tsf-1* (*Figure 1—figure supplement 1C and D*). Taken together, our analyses suggest that the increase in cell number and cell size at the SAM during floral induction induced by LDs is mediated by both the photoperiodic flowering pathway and GA signals, and these appear to act at least partially through different mechanisms.

## A GA biosynthesis enzyme is expressed in the SAM during floral transition and contributes to meristem size at doming

Transcription of the *GA20ox2* gene, which encodes a GA biosynthesis enzyme, is induced at the SAM prior to the floral transition (*Andrés et al., 2014*). However, the detailed spatiotemporal expression pattern of *GA20ox2* has not been analyzed. Therefore, we generated a *VENUS-GA20ox2* line that contains the 6.6 kb promoter and the 201 bp 3′ UTR. The transgene was introduced into the *ga20ox2*–1 mutant and complemented its late-flowering phenotype under SDs (*Figure 2—figure supplement 1I*). The VENUS-GA20ox2 signal was restricted to the abaxial epidermis of leaf primordia in SDs, and expanded into the SAM after plants were transferred to LDs (*Figure 2A–D* and *Figure 2—figure supplement 2A–C*). Notably, the VENUS-GA20ox2 signal was detected not only in the RZ but also in the PZ, predominantly in the L1 layer, during floral transition at 3 and 5 days after transfer to LDs (*Figure 2B and C*). After floral transition, VENUS-GA20ox2 expression was reduced again in the inflorescence meristem at 7 LDs (*Figure 2D* and *Figure 2—figure supplement 2D*). This dynamic change in VENUS-GA20ox2 expression was consistent with the levels of *GA20ox2* mRNA detected by RT-qPCR in the shoot apex of wild-type plants transferred from SDs to LDs (*Figure 2—figure supplement 1H*) and was also observed in plants that were germinated and grown continuously in LDs (*Figure 2—figure supplement 1A–G*), suggesting that it is tightly associated with the developmental transition of the meristem. The expression in the PZ was consistent in all transgenic lines that complemented the late-flowering phenotype of *ga20ox2*–1 and we used *pGA20ox2:: VENUS-GA20ox2 #14* in the following experiments (*Figure 2—figure supplement 1F,G, and I*).

*GA20ox2* transcription is downregulated by excessive amounts of GA via a feedback mechanism (*Rieu et al., 2008b*). Therefore, we examined whether the presence of VENUS-GA20ox2 in the SAM is affected by altering GA levels. Treatment with paclobutrazol (PAC), a GA biosynthesis inhibitor, increased the signal intensity of VENUS-GA20ox2 at all time points, although consistently, no expression was detected in the inner domain of the CZ (*Figure 2I–L* and *Figure 2—figure supplement 2I–L*). Furthermore, after treatment with GA$_3$, the VENUS-GA20ox2 signal was still detected in the PZ of the SAM and was only slightly reduced in signal intensity (*Figure 2E–H* and *Figure 2—figure supplement 2E–H*). These physiological assays indicated that the spatial distribution pattern of VENUS-GA20ox2 in the SAM during floral transition is not significantly affected by endogenous GA levels.

Next, we analyzed the expression of VENUS-GA20ox2 in the L1 layer of the SAM during floral transition at 2wSD+3LD (*Figure 2M*). Consistent with our observation in the longitudinal sections, VENUS-GA20ox2 expression was lower in the CZ and higher in the PZ after exposure to three LDs. Notably, VENUS-GA20ox2 signal was not evenly distributed in the PZ, but was detected in distinct patches (*Figure 2M*). Considering the positions of leaf primordia, the regions that express VENUS-GA20ox2 probably correspond to incipient primordia (*Figure 2M*).

The expression of VENUS-GA20ox2 was also examined under continuous SDs. The signal was weakly detected only in the abaxial side of leaf primordia up to 5wSD, but was hardly detectable from 6wSD, when floral primordia are produced instead of leaf primordia (*Figure 2—figure supplement 3A*). Plants grown under SDs for different time periods were transferred to LDs, and in those plants that were still in the vegetative phase (2wSD, 3wSD, 4wSD, and 5wSD) the VENUS-GA20ox2 signal was upregulated after exposure to 3LDs (*Figure 2—figure supplement 3B*). Notably, the VENUS-GA20ox2 signal was induced in the meristem only in the samples transferred to LDs after 2- or 3wSD. The plants transferred to LDs at later time points exhibited the VENUS-GA20ox2 signal in leaf primordia, but not in the SAM (*Figure 2—figure supplement 3B*).

The stage-specific pattern of VENUS-GA20ox2 expression in the PZ prompted us to examine whether GA20ox2 contributes to the increase in cell number and cell size during floral transition. We compared meristem size in *ga20ox2*–1 and Col-0. In this experiment, the increase in meristem area and cell area was visible in 3LD and 5LD after transfer from SD in both genotypes (*Figure 3A and B*). However, in the 5LD samples, the meristem area was smaller in *ga20ox2*–1 compared with Col-0 (*Figure 3B*). Analysis of different regions of the meristem, as described previously in *Figure 1—figure supplement 2*, at 5LD detected a reduced number of cells and the presence of smaller cells in the ring outside the central region (*Figure 3B–F*). Because the *ga20ox2*–1 mutant is late flowering (*Figure 2—figure supplement 1I*), we monitored these parameters until 9LD after transfer, but the meristem size or cell area in the PZ was not greater at any time point than that observed in Col-0 at

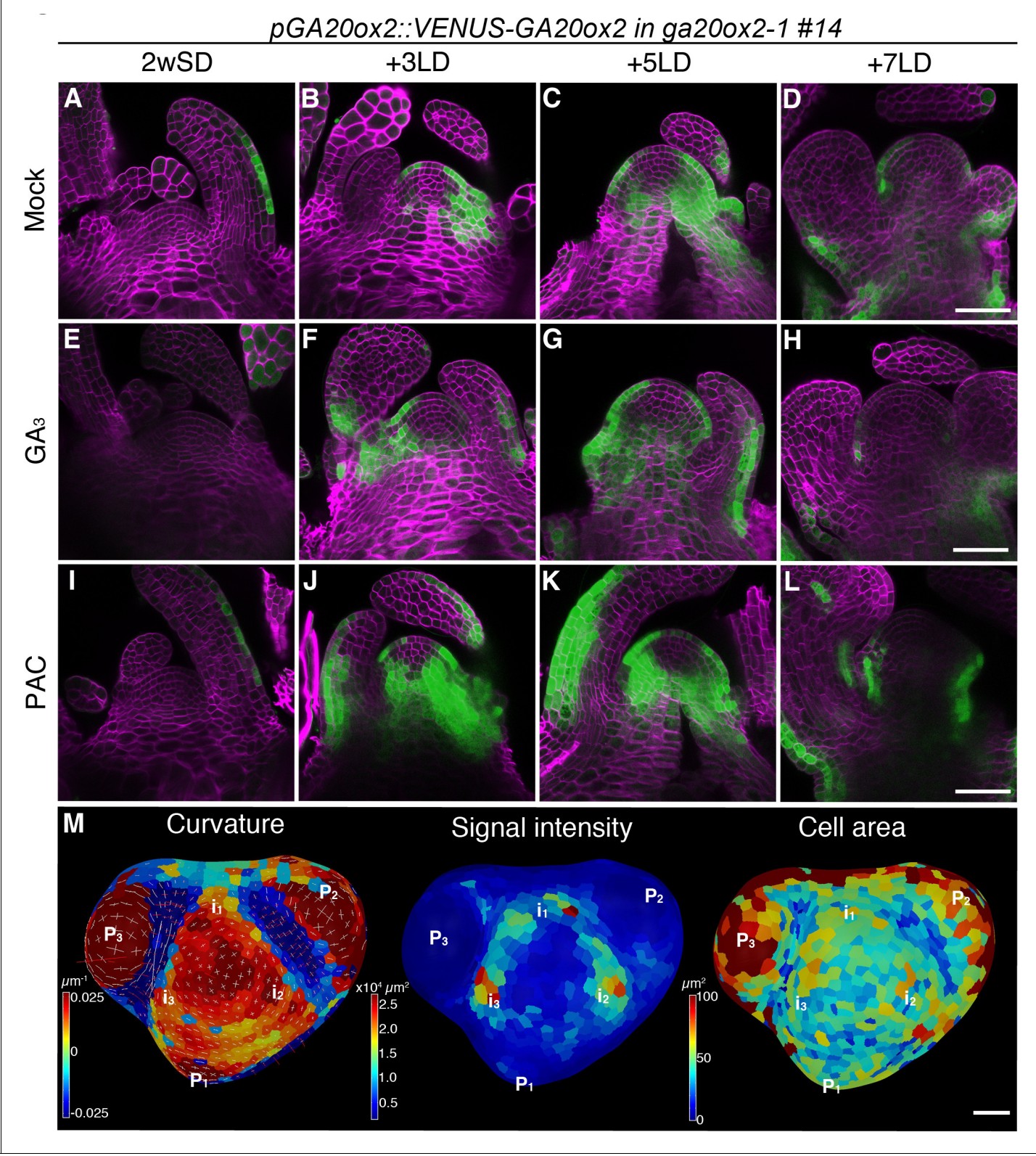

**Figure 2.** The expression pattern of *GA20ox2* in the shoot apical meristem during floral transition. (**A–L**) Confocal imaging of longitudinal sections of a meristem expressing *pGA20ox2::VENUS-GA20ox2* (*ga20ox2*–1 background line #14; green). The plants were germinated and grown on growth medium in short days (SDs) for 2 weeks (**A, E, and I**), then transferred to long days (LDs) for 3LDs (**B, F, and J**), 5LDs (**C, G, and K**), or 7LDs (**D, H, and L**). Samples were treated either with mock (**A–D**; ethanol), 100 µM GA₃ (**E–H**), or 10 µM PAC (**I–L**) for 24 hr prior to harvesting. The cell walls were stained with

*Figure 2 continued on next page*

*Figure 2 continued*

Renaissance 2200 (magenta). (**M**) Segmented surface projection of *pGA20ox2::VENUS-GA20ox2 #14* grown for 2 weeks in SDs and 3LDs (2wSD3LDs). Gaussian curvature (left panel), VENUS signal intensity (middle panel), and cell area (right panel) were extracted. The positions of future primordia ($i_n$) were predicted from those of existing primordia ($P_n$). Scale bars, 50 µm (**A–L**), 20 µm (**M**).

The online version of this article includes the following figure supplement(s) for figure 2:

**Figure supplement 1.** Expression pattern of GA20ox2 during floral transition.
**Figure supplement 2.** The expression pattern of *GA20ox2* in the shoot apical meristem during floral transition.
**Figure supplement 3.** Expression pattern of GA20ox2 under short days.

5LD, although at 9LD cell size was larger in the CZ and PZ of *ga20ox2*–1 meristems than in those of Col-0 plants (*Figure 3E and F*). This suggests that activity of *GA20ox2* contributes to the increase in cell size in the PZ at 5LD during floral transition, but the overall effect of *ga20ox2*–1 on meristematic cell size is weaker than that of *KNAT1:GA2ox7* (*Figures 1G* and *3D*), indicating that other enzymes might also participate in increasing GA level at this stage.

## A GA catabolism enzyme is downregulated in the SAM during floral transition

Endogenous GA levels are regulated by both GA biosynthesis and deactivation (*Yamaguchi, 2008*). Among five $C_{19}$-GA 2-oxidases involved in GA deactivation, GA2ox4 plays the major role in regulating flowering time under SDs, whereas GA2ox2 and GA2ox6 play minor roles (*Rieu et al., 2008a*). To examine the contribution of GA2 oxidases in flowering under LDs, we scored the number of leaves produced before flowering under those conditions. Although the contribution of GA to flowering under LDs was smaller than under SDs, the quintuple mutant in which all five $C_{19}$-*GA2ox* genes are mutated (*ga2ox1*–1, *ga2ox2*–1, *ga2ox3–1*, *ga2ox4*–1, and *ga2ox6*–2) flowered with significantly fewer rosette leaves than wild-type plants under LDs (*Figure 4—figure supplement 1A–C*). Notably, the *ga2ox2–1 ga2ox4–1 ga2ox6*–2 triple mutant and *ga2ox4*–1 single mutant flowered as early as the *ga2ox* quintuple mutant (*Figure 4—figure supplement 1A–C*), suggesting that GA2ox4 is the major enzyme in this class that delays flowering time under LDs. Another GA catabolism enzyme, EUI-LIKE P450 A1 (ELA1), was previously described to promote floral meristem identity by reducing the level of bioactive GA in floral primordia (*Yamaguchi et al., 2014*). Thus, *ela1* mutants produced cauline leaves on the inflorescence stem at nodes that produce flowers in wild-type plants. To examine whether $C_{19}$-GA2ox enzymes are involved in flower formation, we counted the number of cauline leaves produced on the inflorescence stem of *ga2ox* mutants. The number of cauline leaves was indistinguishable in *ga2ox* mutants and wild-type plants (*Figure 4—figure supplement 1D*), suggesting that $C_{19}$-GA2ox enzymes increase the duration of vegetative development prior to shoot elongation, but are not involved in regulating the transition from cauline leaves to floral primordia after bolting. However, *ga20ox2* mutants possessed more rosette leaves and fewer cauline leaves in LDs (*Figure 4—figure supplement 1C and D*), suggesting that in wild-type plants GA20ox2 contributes to the synthesis of GA that promotes transition from the vegetative to inflorescence stage and delays the switch from cauline leaves to floral meristem identity (*Yamaguchi et al., 2014*). These data support previous reports that GA20ox2 is one of the key GA biosynthesis enzymes that contributes to GA production at the shoot apex during floral transition (*Andrés et al., 2014*; *Rieu et al., 2008b*).

Next, we examined the expression of *GA2ox* genes in shoot apices. After transfer of plants from 3wSDs to LDs, the relatively high level of *GA2ox4* mRNA under SDs was downregulated after 3–5 days in LDs, and upregulated again after 7 days in LDs (*Figure 4—figure supplement 1J*). This dynamic change in *GA2ox4* mRNA was almost complementary to that in *GA20ox2* expression (*Figure 2—figure supplement 1I* and *Figure 4—figure supplement 1J*). *GA2ox2* and *GA2ox4* are expressed in shoot apices (*Jasinski et al., 2005*), but their spatiotemporal expression patterns in the SAM during floral transition have not been defined. Analysis of the translational fusion lines showed that VENUS-GA2ox2 and VENUS-GA2ox6 were expressed in the mid-vein of the young leaf primordia and the stipules, respectively, but not in the SAM (*Figure 4—figure supplement 1E–I and O–S*). By contrast, VENUS-GA2ox4 signal was strongly detected in the abaxial side of leaf primordia and in the PZ of the vegetative SAM, where the future primordia will be initiated (*Figure 4A and E*,

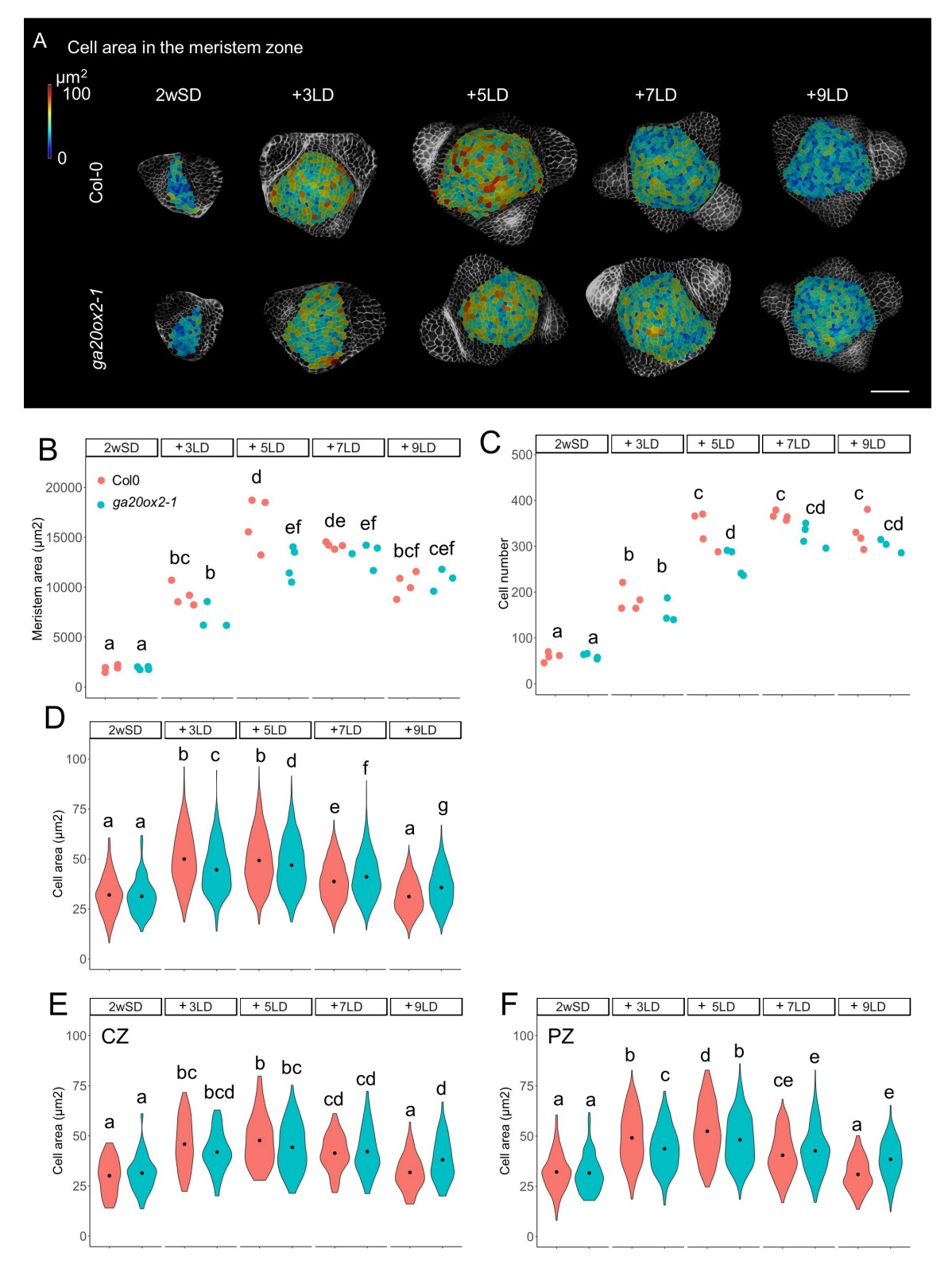

**Figure 3.** GA20ox2 contributes to meristem size at floral transition. (A) Heat-map quantification of cell area in the meristem region of Col-0, and *ga20ox2*–1 grown for 2 weeks in non-inductive short days (SDs) (2wSD) and transferred to inductive long days (LDs) for 3LDs (+3LD), 5LDs (+5LD), 7LDs (+7LD), or 9LDs (+9LD). Scale bars, 50 µm. (B–F) Quantification of the meristem area (B), the cell number (C), and the cell area in the meristem region (D–F) in Col-0, and *ga20ox2*–1 grown for 2 weeks in non-inductive SDs (2wSD) and transferred to inductive LDs for 3LDs (+3LD), 5LDs (+5LD), 7LDs

*Figure 3 continued on next page*

*Figure 3 continued*

(+7LD), or 9LDs (+9LD). Letters a–g in panels B–F show significant differences between conditions and genotypes (p<0.05, using ANOVA followed by Tukey's pairwise multiple comparisons), n = 3–4 apices.

The online version of this article includes the following source data for figure 3:

**Source data 1.** Original data of meristem area and cell number of each genotype for *Figure 3A–C* and *Figure 4—figure supplement 6B and C*.
**Source data 2.** Original data of meristem area and cell number of each genotype for *Figure 3D–F* and *Figure 4—figure supplement 6D–F*.

*Figure 4—figure supplement 1K*, *Figure 4—figure supplement 2A*, *Figure 4—figure supplement 3A and E*, and *Figure 4—figure supplement 4B–E*). Three days after transfer of 2wSD-grown plants to LDs, the VENUS-GA2ox4 signal was excluded from the PZ and restricted to the abaxial side of primordia (*Figure 4B and F*, *Figure 4—figure supplement 1L*, *Figure 4—figure supplement 2B*, and *Figure 4—figure supplement 3B*), and almost disappeared 5 days after transfer (*Figure 4C and G*, *Figure 4—figure supplement 1M*, *Figure 4—figure supplement 2C*, and *Figure 4—figure supplement 3C*). However, 7 days after transfer to LDs, VENUS-GA2ox4 signal was detected again strongly in floral primordia (*Figure 4D and H*, *Figure 4—figure supplement 1N*, *Figure 4—figure supplement 2D*, and *Figure 4—figure supplement 3D*). The *VENUS-GA2ox4* transgene complemented the early-flowering phenotype of *ga2ox4*–3 (*Figure 4—figure supplement 4*), suggesting that the dynamic expression pattern of VENUS-GA2ox4 is necessary and sufficient for the role of *GA2ox4* in flowering (*Figure 4* and *Figure 4—figure supplement 4*). A previous study showed that *GA2ox4* transcript level is upregulated by GA treatment of wild-type seedlings via a feedforward mechanism (*Rieu et al., 2008a*). Indeed, treatment of 2wSD-grown plants with exogenous GA caused expansion of the VENUS-GA2ox4 signal into the RZ, but it was still not detected in the CZ of the SAM (*Figure 4I* and *Figure 4—figure supplement 3I*). By contrast, exogenous GA treatment did not enhance the VENUS-GA2ox4 signal 5 days after transfer to LDs even in the PZ or RZ (*Figure 4K* and *Figure 4—figure supplement 3J–L*). These observations suggest that the feedforward regulation of *GA2ox4* by GA is active in the SAM during the vegetative stage but is blocked during floral transition.

Under continuous SDs, VENUS-GA2ox4 signal was detected strongly in the young leaf primordia and the abaxial side of developing leaves during the vegetative stage (*Figure 4—figure supplement 5A*). The signal intensity was significantly weaker between 4wSD and 5wSD (*Figure 4—figure supplement 5A*), when the floral transition occurs under these conditions (*Hyun et al., 2016*). In the reproductive stage (6 and 7wSD), the strong signal was detected again in the floral primordia (*Figure 4—figure supplement 5A*). These observations suggest that the expression of *GA2ox4* changes dynamically depending on the developmental stage of the floral transition, even under SDs. The signal decreased after exposure to three LDs at all time points, among which 4wSD3LD showed strongest reduction (*Figure 4—figure supplement 5B*). This result again confirmed the tight correlation between *GA2ox4* downregulation and the floral transition.

Furthermore, after transfer of 2wSD-grown plants to LDs, the size of the meristem and the cell number in the SAM of *ga2ox4*–3 were indistinguishable from those of wild-type Col-0 (*Figure 4—figure supplement 6A–C*), consistent with *GA2ox4* expression being reduced in Col-0 plants under these conditions. Nevertheless, a significant difference in cell area was observed between Col-0 and *ga2ox4*–3 SAMs after exposure to 5 LDs, with the cells of *ga2ox4*–3 being slightly larger than those of Col-0 (*Figure 4—figure supplement 6D*). This difference was not observed within five cells from the center, suggesting that *GA2ox4* modulates the size of cells in the PZ closer to the boundary region (*Figure 4—figure supplement 6E and F*).

## After floral transition, GA2ox4 accumulates in young floral primordia

A previous study demonstrated that reduced GA levels in young floral primordia allow RGA to accumulate and interact with SPL proteins to promote *AP1* transcription and confer floral identity (*Yamaguchi et al., 2014*). To examine whether GA2ox4 may contribute to reducing GA levels during the acquisition of floral identity, we compared its spatial expression pattern with that of AP1 during floral transition and in the mature inflorescence (*Figure 5*). In the mature inflorescence meristem at 18 LDs, VENUS-GA2ox4 signal was detected in young floral primordia that did not yet express AP1-GFP and in older primordia that did express AP1-GFP (*Figure 5C,D and G–H*). This observation

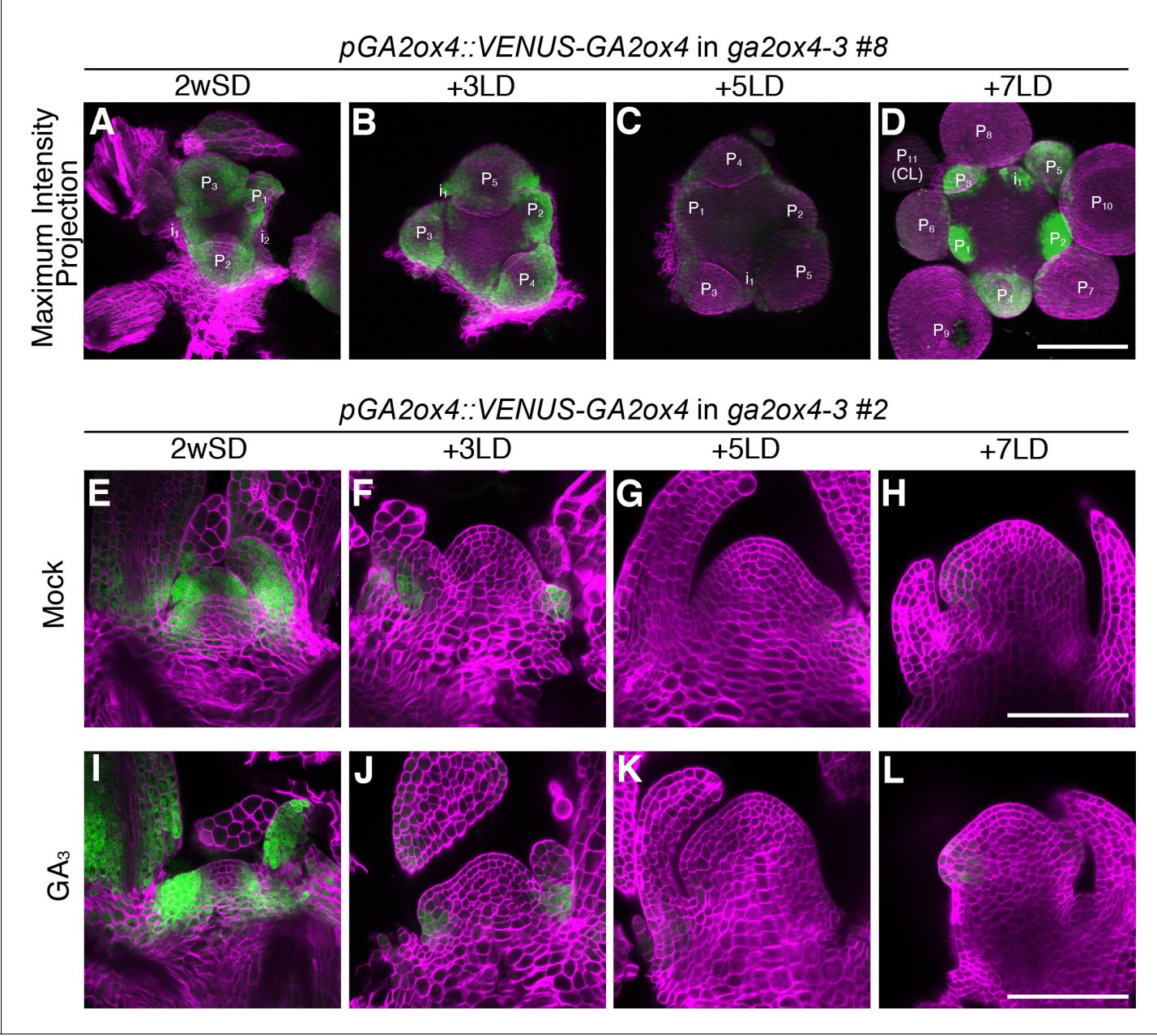

**Figure 4.** The expression pattern of GA2ox4 in the shoot apical meristem during floral transition. (A–D) Maximum intensity projection images of the meristems in *GA2ox4::VENUS-GA2ox4 ga2ox4*–3 line #8 (green), grown for 2 weeks in short days (SDs) (A) and then transferred to long days (LDs) for 3 (B), 5 (C), or 7 (D) days. (E–L) Longitudinal confocal images of *GA2ox4::VENUS-GA2ox4 ga2ox4*–3 line #2 (green) treated without (E–H) or with (I–L) 100 µM GA$_3$ for 24 hr prior to harvesting. The plants were germinated and grown on growth medium for 2 weeks in SDs (E and I), and then transferred to LDs for 3 (F and J), 5 (G and K), or 7 (H and L) days. Cell walls were stained with Renaissance 2200 (magenta). Scale bars, 100 µm.

The online version of this article includes the following figure supplement(s) for figure 4:

**Figure supplement 1.** Contribution of C$_{19}$-GA2oxs to flowering time.
**Figure supplement 2.** Expression pattern of VENUS-GA2ox4 during floral transition.
**Figure supplement 3.** The expression pattern of GA2ox4 in the shoot apical meristem during floral transition.
**Figure supplement 4.** Expression pattern of VENUS-GA2ox4 in independent transformants.
**Figure supplement 5.** Expression pattern of GA2ox4 under continuous short days.
**Figure supplement 6.** The number and size of cells in the meristem in *ga2ox4*.

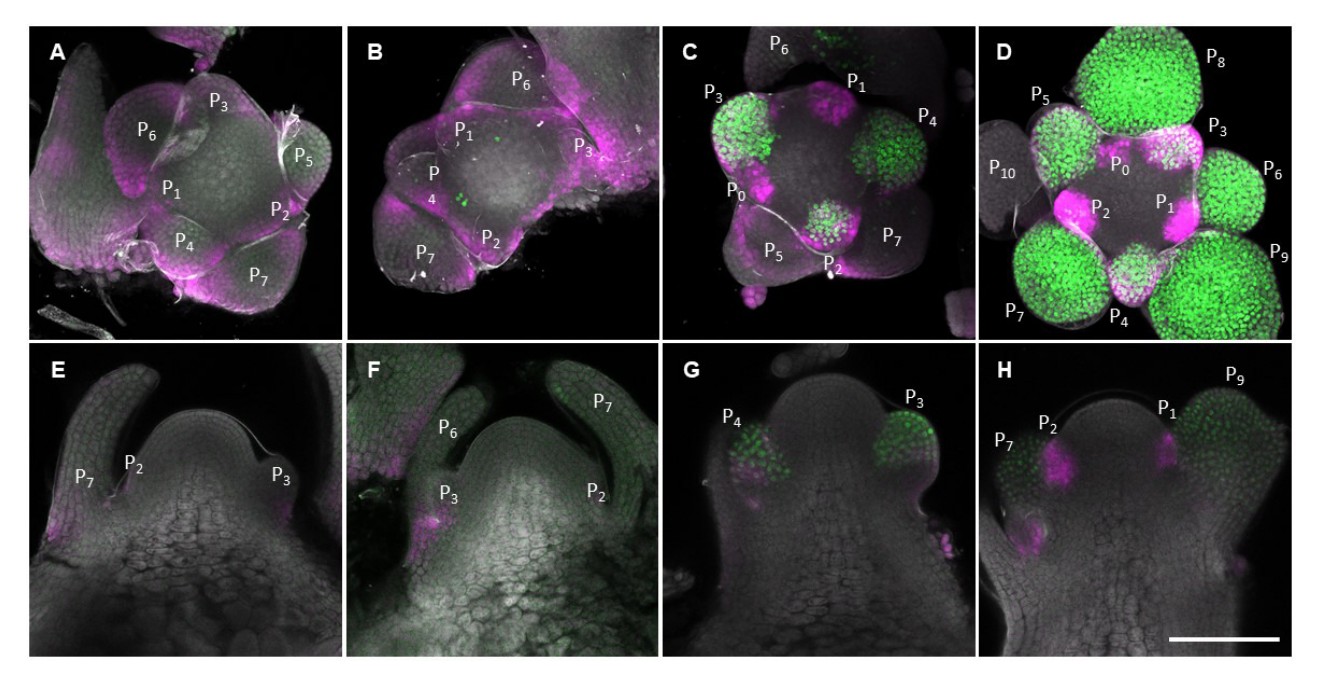

**Figure 5.** Spatial expression patterns of AP1 and GA2ox4 during floral transition. (**A–H**) Expression of *AP1::AP1-GFP* (green) and *GA2ox4::VENUS-GA2ox4 #8* (Magenta) grown under long days for 14 (**A, B, E, and F**) and 18 (**C, D, G, and H**) days. Maximum intensity projections (**A–D**) and longitudinal sections are shown. Scale bar, 100 µm.

suggests that *GA2ox4* contributes to the reduction in GA levels in the floral primordia of the mature inflorescence. However, VENUS-GA2ox4 signal was hardly detectable in the young primordia during floral transition at 14 LDs (*Figure 5A,B,E, and F*), suggesting that at this transition stage GA2ox4 may not strongly contribute to the deactivation of GA to induce *AP1* expression. This observation was consistent with our phenotyping analysis, which showed that the number of cauline leaves did not increase in *ga2ox4* mutants, suggesting that acquisition of floral meristem identity was not impaired (*Figure 4—figure supplement 1D*). The AP1-GFP signal was occasionally detected in developed leaf primordia prior to its expression in floral primordia, which is probably due to the *AP1* promoter activity, as reported previously (*Hempel et al., 1997*).

## The transcription of *GA20ox2* and *GA2ox4* is regulated by SOC1 and SVP

The dynamic expression patterns of *GA20ox2* and *GA2ox4* tightly correlated with exposure to LDs and the developmental stage of the plants. Therefore, we examined the involvement of photoperiodic flowering pathway genes on the expression patterns of *GA20ox2* and *GA2ox4*. Previously, we reported that *GA20ox2* was indirectly regulated by SVP in SDs (*Andrés et al., 2014*). However, the expression of *GA20ox2* under inductive LDs has not been examined. The transcript level of *GA20ox2* was therefore tested by RT-qPCR in shoot apices from 9 to 19 days after germination under LDs (*Figure 6A* and *Figure 6—figure supplement 1A*). The level of *GA20ox2* transcript was not affected in the *ft-10 tsf-1* mutant background, but was greatly elevated in *svp-41* mutant apices compared to Col-0 at 9 and 11 days after germination (*Figure 6A* and *Figure 6—figure supplement 1A*). SVP has antagonistic functions to SOC1 during floral transition (*Hartmann et al., 2000*; *Jang et al., 2009*), and redundant roles during early floral primordia development (*Gregis et al., 2009*; *Liu et al., 2009*), so we also tested *GA20ox2* mRNA levels in *soc1-2* mutants. *GA20ox2* transcript level was higher in *soc1-2* mutants than in Col-0 at later time points (*Figure 6B*). The increase in *GA20ox2* mRNA in *soc1-2* is probably not due to altered *SVP* expression, because the *soc1* mutation caused only a minor effect on *SVP* mRNA level under LDs at these time points (*Figure 6—figure*

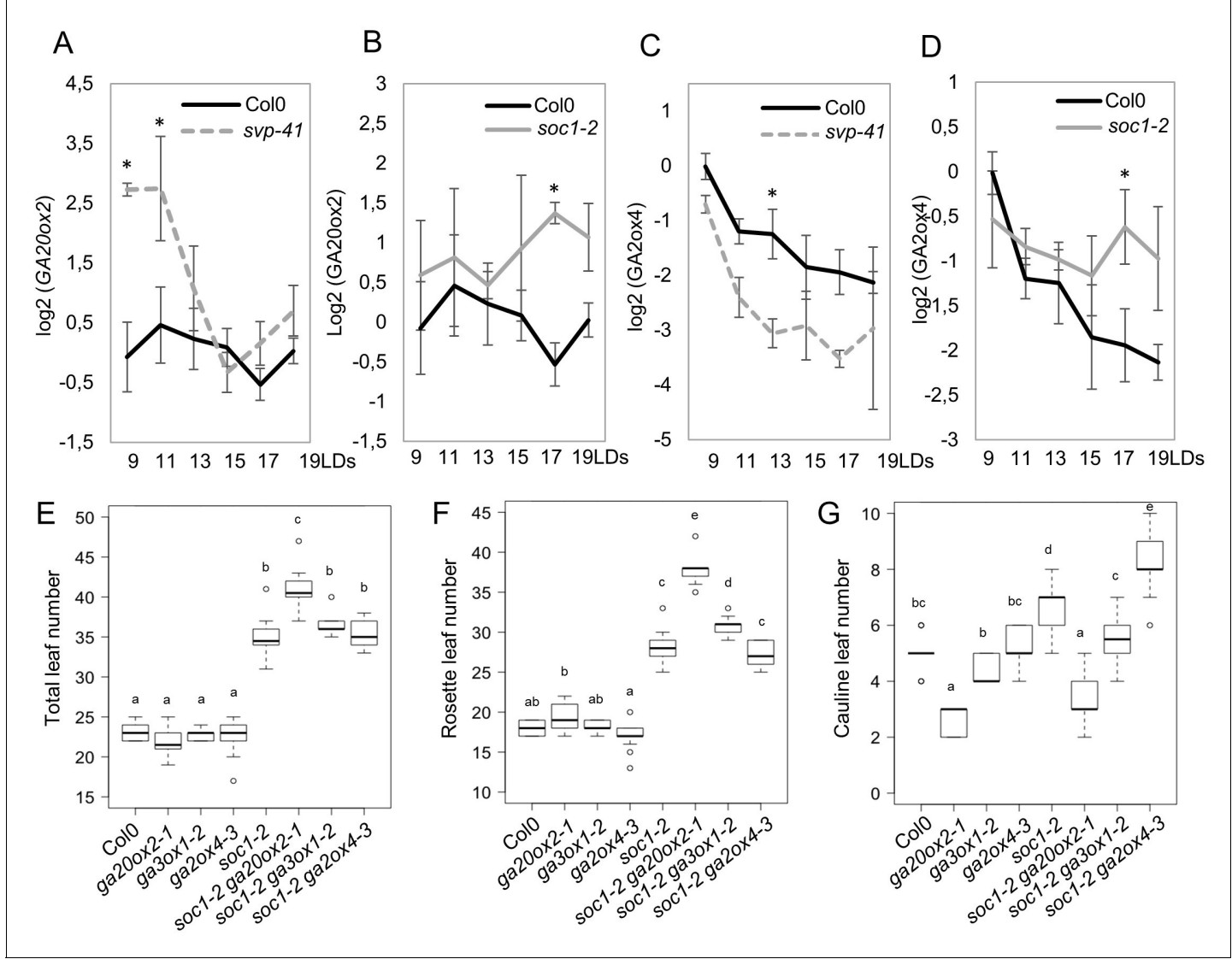

**Figure 6.** Regulation of *GA20ox2* and *GA2ox4* by SOC1 and SHORT VEGETATIVE PHASE (SVP). (**A and B**) Temporal expression pattern of *GA20ox2* mRNA in apices of wild type (**A and B**), *svp-41* (**A**), and *soc1-2* (**B**). (**C and D**) Temporal expression pattern of *GA2ox4* mRNA in apices of wild type (**C and D**), *svp-41* (**C**), and *soc1-2* (**D**). All samples were harvested 8 hr after dawn. Asterisks show significant differences between conditions in the comparisons indicated (p<0.05, using ANOVA followed by Tukey's pairwise multiple comparisons). (**E–G**) Genetic interaction tests between mutants of GA metabolism genes and *soc1*. The number of total leaves (**E**), rosette leaves (**F**), and cauline leaves (**G**) were scored for wild type (Col-0), *ga20ox2*–1, *ga3ox1*–2, *ga2ox4*–3, *soc1-2*, *soc1-2 ga20ox2*–1, *soc1-2 ga3ox1*–2, and *soc1-2 ga2ox4*–3 (n ≥ 13; a–e indicate significant differences calculated with ANOVA, Tukey's Honest Significant Difference (HSD) test; p<0.001).

The online version of this article includes the following source data and figure supplement(s) for figure 6:

**Source data 1.** Original RT-qPCR data of different genotypes for *Figure 6A–D* and *Figure 6—figure supplement 1*.

**Source data 2.** Original data of leaf number of different genotypes for *Figure 6E–G*.

**Figure supplement 1.** Effect of photoperiodic mutants on transcripts of gibberellin (GA) metabolism genes and their genetic interaction.

*supplement 1C*). These results suggest that SVP and SOC1 downregulate *GA20ox2* at different developmental stages.

We also used MorphoGraphX to analyze the morphology of the shoot meristem of *soc1-2* mutants during the floral transition to compare this with the previous analysis of *ga20ox2*–1 and *ga2ox4*–3 mutants (*Figure 3* and *Figure 4—figure supplement 6*). As performed previously, SD-grown plants were transferred to LDs and the SAM analyzed. The meristem area and mean cell size in the meristem of the *soc1-2* mutant were smaller than those of Col-0 at +5LD (*Figure 7A,B, and*

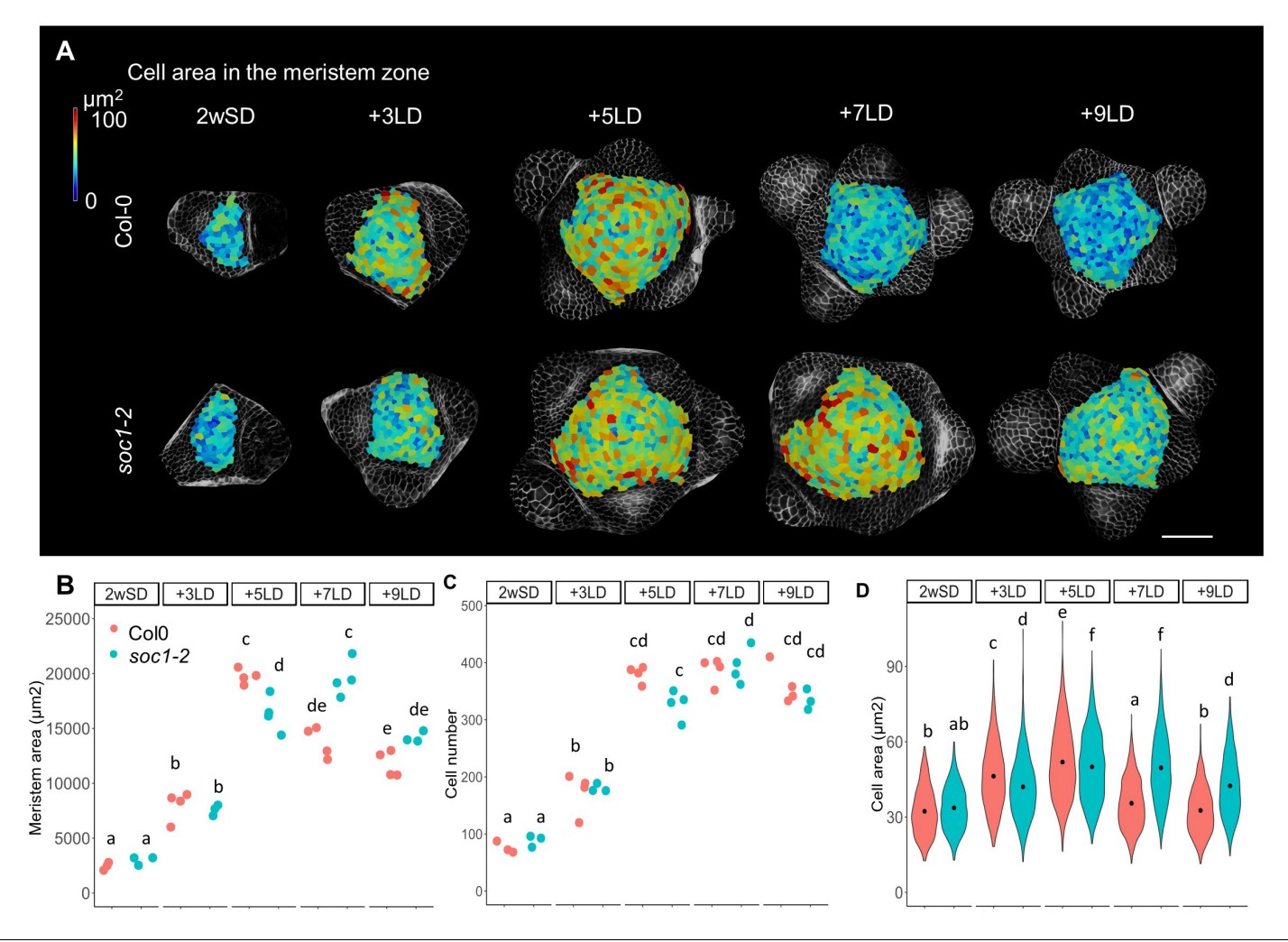

**Figure 7.** Protracted doming of the *soc1* mutant in response to long days (LDs). (**A**) Heat-map quantification of cell area in the meristem region of Col-0, and *soc1-2* grown for 2 weeks in non-inductive short days (SDs) (2wSD) and transferred to inductive LDs for 3LDs (+3LD), 5LDs (+5LD), 7LDs (+7LD), or 9LDs (+9LD). Scale bars, 50 µm. (**B–D**) Quantification of the meristem area (**B**), the cell number (**C**), and the cell area in the meristem region (**D**) in Col-0, and *soc1-2* grown for 2 weeks in non-inductive SDs (2wSD) and transferred to inductive LDs for 3LDs (+3LD), 5LDs (+5LD), 7LDs (+7LD), or 9LDs (+9LD). Letters a–f in panels B–F show significant differences between conditions and genotypes (p<0.05, using ANOVA followed by Tukey's pairwise multiple comparisons), n = 3–4 apices.

The online version of this article includes the following source data for figure 7:

**Source data 1.** Original data of meristem area and cell number of each genotype for *Figure 7B and C*.
**Source data 2.** Original data of cell size of each genotype for *Figure 7D*.

*D*). However, the *soc1-2* mutant SAM continued to increase in size and at +7LD the meristem area and cell number of the mutant were similar to those of Col-0 at +5LD (*Figure 7B and C*). This result is consistent with the late-flowering phenotype of the *soc1-2* mutant and suggests that the SAM of *soc1-2* mutant domes to the same extent as Col-0, but that the process is more gradual and extends over 2 days longer. This result is in contrast to that observed for *ft tsf* mutants, which did not form a domed meristem throughout the time course (*Figure 1*).

We then examined the effect of the *ga20ox2* mutation on the flowering time of *soc1* mutants. *SOC1* is a positive regulator of flowering that functions downstream of *FT* and *TSF*, and *soc1* mutation results in late flowering under LDs (*Samach et al., 2000*). Consistent with these reports, under LD conditions *soc1* produced 57% more rosette leaves and 36% more cauline leaves than wild-type plants (*Figure 6E–G*). By contrast, *ga20ox2* showed only a minor delay in flowering time compared

to wild-type plants (6% more rosette leaves), but it delayed flowering of *soc1* in a synergistic manner (33% more rosette leaves than *soc1-2*; *Figure 6B,E, and F*). Notably, the number of cauline leaves was lower in the *soc1 ga20ox2* double mutant and was almost similar to that in *ga20ox2* (p>0.1), suggesting that the increase in the number of cauline leaves in *soc1* was mainly caused by the upregulation of *GA20ox2* (*Figure 6G*), consistent with the idea that high GA levels repress floral meristem identity (*Yamaguchi et al., 2014*). The synergistic effect of *ga20ox2* on *soc1* is not simply caused by the general growth defect of GA deficiency, because mutation in another GA biosynthesis enzyme gene, *GA3ox1*, had only a minor effect on *soc1* flowering time (*Figure 6E–G*).

We also examined the effect of genes in the photoperiodic flowering pathway on the mRNA level of *GA2ox4*. In *soc1*, higher *GA2ox4* mRNA levels were maintained for several days, suggesting that *SOC1* can negatively regulate *GA2ox4* expression (*Figure 6D*). Although *ga2ox4* mutation resulted in slightly earlier flowering than wild-type plants (6% fewer rosette leaves), it did not significantly affect the late-flowering phenotype of *soc1-2* (4% fewer rosette leaves than *soc1-2*) (*Figure 6F*). However, the number of cauline leaves was significantly higher in *soc1-2 ga2ox4–3* compared to *soc1-2* (17% more cauline leaves than *soc1-2*), suggesting that increased expression of *GA2ox4* delayed flower formation in *soc1-2* during later developmental stages (*Figure 6G*). Again, the level of *GA2ox4* mRNA was not affected in *ft-10 tsf-1* (*Figure 6—figure supplement 1B*). These data suggest that *SOC1* is one of the factors that regulate the expression of *GA20ox2* and *GA2ox4* downstream of *ft-10 tsf-1*. Indeed, the *ft-10 tsf-1* double mutant flowered much later than *soc1-2*, and the *ga2ox4* mutation slightly decreased the rosette leaf number but did not affect the cauline leaf number of *ft-10 tsf-1* (*Figure 6—figure supplement 1E–G*).

Previously, no direct binding of SOC1 or SVP to *GA20ox2* was detected by ChIP (*Andrés et al., 2014*; *Immink et al., 2012*; *Mateos et al., 2015*; *Tao et al., 2012*). However, we detected binding of SOC1 to a distal region of the *GA2ox4* promoter that contains several CArG-boxes by ChIP-PCR (*Figure 8A and B* and *Supplementary file 3*). Furthermore, Dexamethasone (DEX)-induced translocation of SOC1:GR into the nucleus in *35S::SOC1:GR* plants decreased the transcript level of *GA2ox4* within 60 min after treatment (*Figure 8D*). These data indicate that SOC1 is a direct negative regulator of *GA2ox4* expression by binding to its 5′ region.

SOC1 and SVP are MADS-domain type transcription factors that often regulate the same target genes (*Tao et al., 2012*). Therefore, we examined the role of *SVP* on *GA2ox4* expression. In loss-of-function *svp-41* mutants, the transcript level of *GA2ox4* was lower than that in wild-type plants at several later stages (*Figure 6C*). These RT-qPCR analyses suggest that *SVP* is a positive regulator of *GA2ox4*, although this might be an indirect effect, because the level of *SOC1* mRNA is increased in *svp-41* (*Figure 6—figure supplement 1D*). Therefore, we examined the binding of SVP to the *GA2ox4* locus by ChIP-qPCR and found that SVP binds to the same region at the distal 5′ end of *GA2ox4* as SOC1. Binding of SVP to the *GA2ox4* locus was increased in *soc1-2*, and an additional binding site within the first intron of *GA2ox4* was detected in this mutant background (*Figure 8C*). These analyses suggest that SVP is a direct, positive regulator of *GA2ox4*, and its binding to target sites is attenuated by SOC1.

Previous histological analyses showed that *SVP* is expressed in the vegetative SAM and floral primordia and therefore overlaps with *GA2ox4* expression (*Hartmann et al., 2000*). Therefore, we examined the tissue in which *SVP* regulates *GA2ox4*, by introducing the *svp-41* mutation into the *GA2ox4::VENUS-GA2ox4* transgenic line. VENUS-GA2ox4 was detected in the SAM of *svp-41* as strongly as in wild-type plants during the vegetative and floral transition stages (*Figure 9A–H* and *Figure 9—figure supplement 1A,B,I, and J*). However, VENUS-GA2ox4 expression was significantly reduced in the floral primordia of *svp-41* during the reproductive stage (*Figure 9D and H* and *Figure 9—figure supplement 1C–H,K, and L*). Indeed, co-localization of VENUS-GA2ox4 signal with SVP-GFP was observed in the floral primordia of wild-type plants carrying both transgenes (*Figure 9I–K* and *Figure 9—figure supplement 1M and N*). These observations suggest that *SVP* is a major positive regulator of *GA2ox4* expression in floral primordia during the reproductive stage.

## Discussion

We describe dynamic changes in shape and cellular composition of the SAM that occur during floral transition and provide insight into how these are controlled. By reconstructing the 2.5D geometry and structure of the SAM from confocal 2D images, we demonstrated that both cell number and

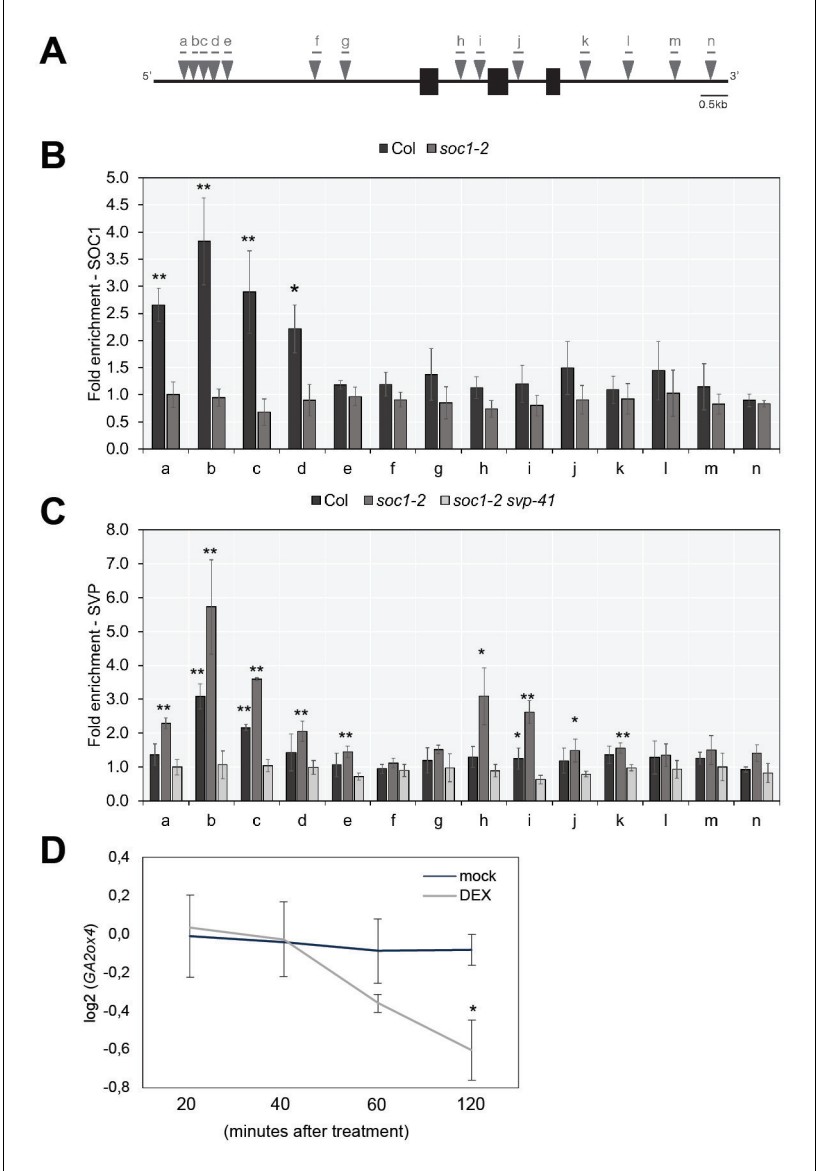

**Figure 8.** *GA2ox4* is directly regulated by SOC1 and SHORT VEGETATIVE PHASE (SVP). (**A**) Schematic representation of the *GA2ox4* locus. Exons are represented by black boxes, introns by white boxes, and UTRs are represented as gray boxes. Consensus binding sites (CArG-boxes) are indicated as triangles. (**B**) ChIP analysis of SOC1 binding to the CArG-boxes at the *GA2ox4* locus. (**C**) ChIP analysis of SVP binding to the CArG-boxes at the *GA2ox4* locus (mean ± SD, *p<0.05, **p<0.01, paired *t*-test). (**D**) Transcript level of *GA2ox4* in *35S::SOC1:GR* plants after mock or DEX treatment. An asterisk shows significant differences between conditions at the indicated time points (p<0.05, using ANOVA followed by Tukey's pairwise multiple comparisons).

The online version of this article includes the following source data for figure 8:

**Source data 1.** Original ChIP-PCR data for *Figure 8*.

size are increased at specific developmental stages in response to inductive environmental cues. The photoperiodic flowering pathway and GA metabolism contribute to these dynamic changes. Genetic analyses demonstrated that two MADS-box transcription factors, SOC1 and SVP, which are components of the gene regulatory network (GRN) associated with floral induction, modulate expression of enzymes involved in GA homeostasis in the SAM and that SOC1 also contributes to the increases in meristematic cell size and number during floral transition. These results highlight the importance of coordinating the photoperiodic and GA phytohormone pathways to induce precise morphological changes in the SAM as it transitions from a vegetative to an inflorescence meristem.

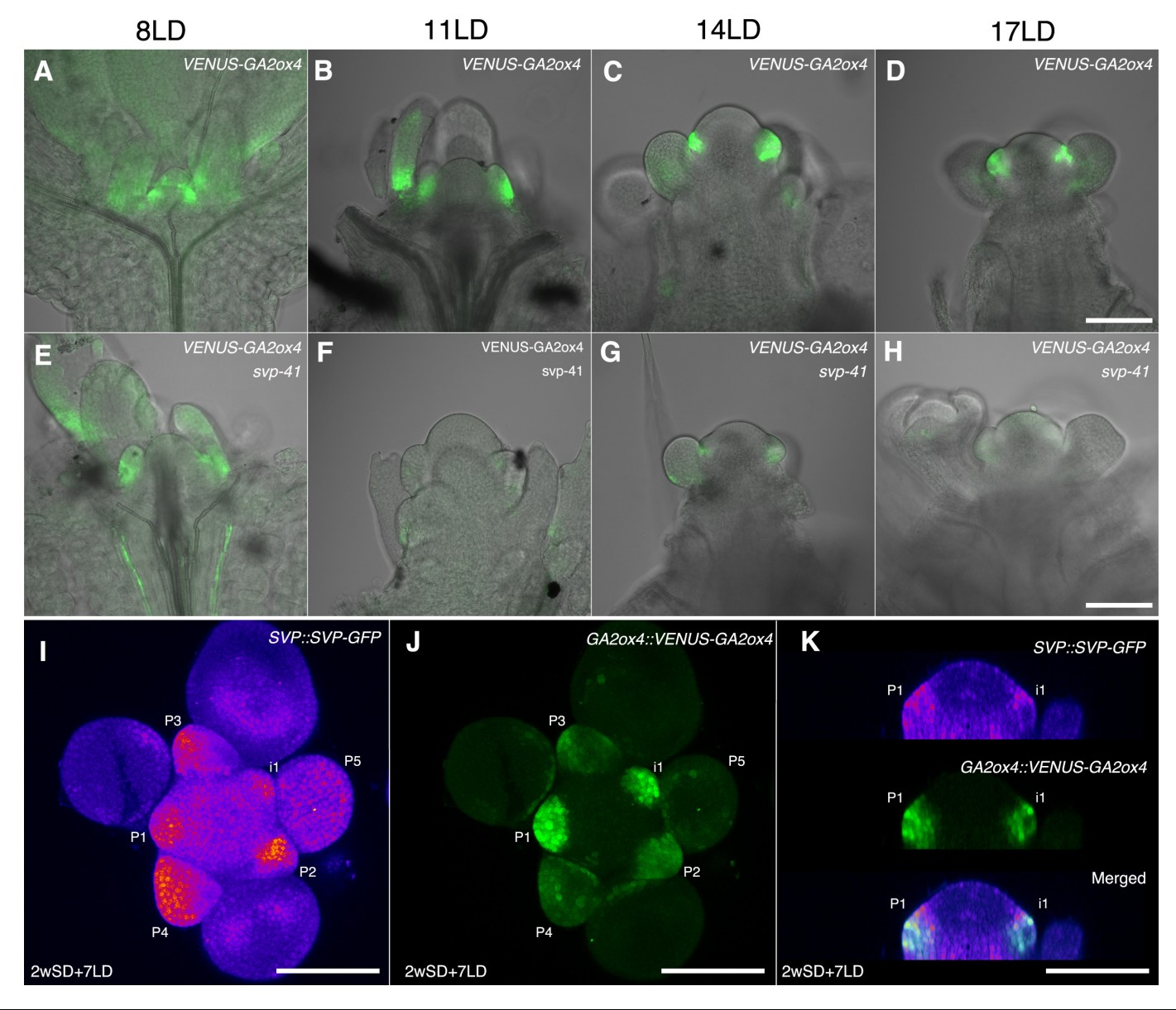

**Figure 9.** SHORT VEGETATIVE PHASE (SVP) regulates *GA2ox4* expression in the floral primordia. (A–H) Confocal imaging of longitudinal sections of meristems expressing *pGA2ox4::VENUS-GA2ox4* in the wild-type (A–D) and *svp-41* (E–H) backgrounds. VENUS-GA2ox4 is detected at a low level in the floral primordia of *svp-41* after floral transition (F–H). (I–K) Expression of *SVP::SVP-GFP* (heat map) and *GA2ox4::VENUS-GA2ox4 #8* (green) grown for 2 weeks in short days (SDs) and transferred to long day for 7 days. Maximum intensity projection (I and J) and optical sections (thickness 0.69 µm, K) are shown. Scale bar, 100 µm. At least four meristems were observed.

The online version of this article includes the following figure supplement(s) for figure 9:

**Figure supplement 1.** SHORT VEGETATIVE PHASE (SVP) enhances *GA2ox4* expression in the floral primordia.

**Figure supplement 2.** Working model summarizing the dynamics of GA2ox4 and GA20ox2 expression in the shoot apical meristem (SAM) during floral transition.

## Environmentally induced changes in shape and cellular composition at the SAM during floral transition

Five days after transfer to long photoperiods, the SAM acquired a characteristic domed shape similar to that previously described for a transition meristem (*Figure 1—figure supplement 1*; *Jacqmard et al., 2003*). The meristematic area at this stage was approximately fourfold greater than

that of plants maintained under SDs, and this enlargement was associated with a threefold to fourfold increase in cell number in the epidermis and an increase of approximately 80% in the median cell area (*Figure 1*). The higher cell number is consistent with the previously described increase in cell division during floral transition (*Bodson, 1975*; *Jacqmard et al., 2003*; *Marc and Palmer, 1982*). Increases in cell size were not previously described in the domed meristem. However, analysis of cell size in the mature inflorescence meristem indicated that it is an emergent property of altering relative growth rate and the rate of cell division (*Jones et al., 2017*). Increases in cell size and cell number in the domed meristem are therefore probably achieved by increasing the rate of both cellular growth and division. During the interval between 5 LDs and 7 LDs, the SAM progressed from a domed transition meristem to a mature inflorescence meristem, and at the latter time point bore on its flanks several floral primordia. During this 2-day interval, the domed shape of the meristem was reduced (*Figure 1—figure supplement 1*) and the meristematic area became smaller, although it remained markedly larger than the vegetative meristem. The reduction in size of the meristematic region at 7 LDs is associated with the presence of primordia closer to the tip of the meristem than occurs in the domed meristem at 5 LDs. This suggests that during the domed transition stage, primordium formation is transiently suppressed, contributing to the larger domed meristematic region. The development of methods for live imaging a single meristem throughout floral induction will be necessary to address this question directly. Between 5 LDs and 7 LDs, a reduction in cell size occurs despite the maintenance of approximately the same number of cells, and this is probably due to a reduction in cell growth rate combined with continued division, as described previously for reduction in cell size in the mature inflorescence meristem (*Jones et al., 2017*). Overall, during the 7-day period after transfer from SDs to LDs, the meristematic area increases as the vegetative meristem becomes an inflorescence meristem that contains more cells of similar size compared to the vegetative meristem. The domed meristem represents an intermediate stage that contains more and larger cells than the vegetative meristem.

Floral induction was induced by transferring plants from SDs to LDs, and this change in environment could conceivably alter meristem size independently of floral induction. Previous work showed that transfer of plants from high to low light intensity caused the inflorescence meristem to become smaller (*Gaillochet et al., 2017*; *Jones et al., 2017*), and the inflorescence meristem of plants grown under nutrient-limiting conditions was smaller than that of plants grown under optimal growth conditions (*Landrein et al., 2018*). The effect of low light was postulated to be due to a reduction in photosynthates; thus, under our conditions, the availability of higher amounts of fixed carbon upon transfer to LDs may contribute to the changes in the SAM that we observed. To control for effects independent of floral transition, *ft-10 tsf-1* plants that did not flower in response to LDs were analyzed (*Figure 1—figure supplement 1* and *Figure 1*). These plants did not produce floral primordia or a domed meristem during the experiment. Thus, the rapid increase in cell number and size associated with doming requires activity of the photoperiodic flowering pathway and are not simply caused by an increase in photosynthesis. However, meristem size, cell number, and cell size did increase gradually in the *ft-10 tsf-1* mutant through 5 and 7 LDs compared with control plants grown under SDs. This residual increase in meristem size in *ft-10 tsf-1* was independent of floral transition, and may be caused by partial activation of the photoperiodic flowering pathway independently of FT and TSF, or to enhanced photosynthesis under longer photoperiods. In further agreement with this notion, specific changes in gene expression were previously detected in the SAM of *ft-10 tsf-1* mutants transferred from SDs to LDs (*Torti et al., 2012*).

In addition to photoperiodic pathway components, GA is probably an important regulator of the increase in SAM size upon floral induction, because after transfer to LDs, the SAM of *KNAT1::GA2o × 7* plants contained fewer cells than wild-type meristems and a similar number to *ft-10 tsf-1* meristems. The *KNAT1* promoter is not active in the epidermis (*Lincoln et al., 1994*; *Porri et al., 2012*), so reducing GA levels in the inner layers of the SAM must lead to the reductions in cell size and cell number detected in the epidermis, perhaps due to indirectly reducing GA levels in the epidermis or to mechanical constraints. Whether the photoperiod and GA pathways regulate SAM size independently remains unclear; however, the distinct morphology of the SAM in these two genotypes suggests independent roles for these two signaling pathways.

## Regulation of GA biosynthesis and catabolism at the SAM during floral transition

Low GA levels maintain the undifferentiated state of the SAM and contribute to the boundary between the SAM and elongating cells in differentiating shoot tissues (*Bolduc and Hake, 2009*; *Hay et al., 2002*; *Sakamoto et al., 2001a*). Class I KNOX transcription factors of several species define SAM identity and reduce GA levels in the SAM by regulating transcription of genes encoding GA-metabolic enzymes (*Bolduc and Hake, 2009*; *Hay et al., 2002*; *Jasinski et al., 2005*; *Sakamoto et al., 2001a*; *Sakamoto et al., 2001b*). By contrast, GA promotes flowering in Arabidopsis and during this process acts at the shoot apex as well as in the vasculature (*Bao et al., 2020*; *Galvão et al., 2012*; *Porri et al., 2012*). Furthermore, an increased level of the DELLA protein GAI in the inflorescence meristem leads to a smaller meristem that contains fewer cells, due to the increased expression of a member of the Kip-related protein (KRP) family of cyclin-dependent kinase inhibitors (*Serrano-Mislata et al., 2017*). This suggests that in the wild-type inflorescence meristem, GA may promote cell division by inducing degradation of GAI and thereby reducing KRP levels. It is not yet clear how the negative regulation of bioactive GA by KNOX transcription factors in the SAM is reconciled with increased GA levels at the shoot apex during floral transition and in inflorescence meristems. Although transcription of *GA20ox2* and the accumulation of bioactive $GA_4$ are induced in the shoot apex during floral transition, its precise pattern of expression in the SAM has not been examined (*Andrés et al., 2014*; *Eriksson et al., 2006*). We defined the spatial and temporal expression dynamics of *GA20ox2* in the SAM during floral transition and observed that it was transiently induced in the future organ primordia in the PZ, but not in the CZ (*Figure 9—figure supplement 2*). *KNOX* genes are highly expressed in the CZ and may repress *GA20ox2* in this region even during floral transition, whereas the induction of *GA20ox2* in the PZ must occur by mechanisms associated specifically with floral transition. Furthermore, transcription of *GA2ox4*, which encodes a GA catabolism enzyme, is reduced at the domed stage but highly upregulated later specifically in early-stage floral primordia in the inflorescence meristem. Although KNOX proteins have a general role in reducing GA levels in meristems (*Bolduc and Hake, 2009*; *Hay et al., 2002*; *Jasinski et al., 2005*; *Sakamoto et al., 2001a*; *Sakamoto et al., 2001b*), they are unlikely to have a regulatory function in modulating GA levels in the SAM specifically during floral transition.

Previous studies showed that GA is critical for cell division and elongation in root meristems (*Achard et al., 2009*; *Ubeda-Tomás et al., 2009*; *Ubeda-Tomás et al., 2008*). We found that GA also regulates cell division and elongation in the SAM during floral transition. Furthermore, the patterns of expression of *GA2ox4* and *GA20ox2* suggest that the GA level mainly increases in the PZ of the SAM during floral transition (*Figure 9—figure supplement 2*). Notably, cell size also increased during floral transition in the PZ of wild-type plants, but not in that of *KNAT1::GA2ox7*. Furthermore, the highest level of expression of *GA20ox2* in the PZ was observed in meristems of 2w- or 3w-old SD-grown plants transferred to 3LDs (*Figure 2—figure supplement 3*). This pattern emphasizes the idea that GA may promote enlargement of the relatively small vegetative SAM of these SD grown plants as they initiate floral transition by enhancing cell division and elongation. Subsequent reduction in *GA20ox2* mRNA levels required the SOC1 MADS box transcription factor, as *soc1* mutants expressed higher levels of *GA20ox2* mRNA for longer than Col-0 (*Figure 6B*). Doming of the SAM of the *soc1* mutant also occurred more slowly than that of Col-0 plants and continued for several more days (*Figure 7*). The longer duration of expression of GA20ox2 in the SAM of the *soc1* mutant may contribute to its extended period of doming. Mutations in *GA20ox2* did not reduce the number and size of the cells in the SAM as much as *KNAT1:GA2ox7* so other isoforms of GA20ox may also redundantly contribute to the biosynthesis of bioactive GA during floral transition.

In addition to the elevated levels of GA that are necessary to terminate the vegetative phase and initiate inflorescence development, a reduction in GA levels in primordia is required to initiate floral development (*Yamaguchi et al., 2014*). The dynamic changes in *GA20ox2* and *GA2ox4* expression that we report are consistent with this model and further support the idea that GA is transiently upregulated in the SAM during floral transition, and then later reduced in floral primordia. *GA20ox2* was expressed in the meristem transiently after exposure to 3–5 LDs during the transition, whereas *GA2ox4* was repressed at this stage, but strongly induced in floral primordia at 7 LDs (*Figure 9—figure supplement 2*). At this stage, expression of GA2ox4 was detectable in incipient floral primordia (i1 stage), which precedes *AP1* expression (*Figure 5*). Depletion of GA in these incipient floral

primordia is proposed to be required for the accumulation of the DELLA protein RGA, which then acts as a transcriptional co-activator of SPL9 in the transcription of *AP1* (*Yamaguchi et al., 2014*). The expression pattern of *VENUS-GA2ox4* in incipient floral primordia is consistent with its role in contributing to GA depletion prior to *AP1* transcription (*Figure 5*). Here it would enhance the role of ELA1 enzymes that were previously shown to be expressed in floral primordia to reduce bioactive GA levels (*Yamaguchi et al., 2014*), although it remains uncertain whether ELA1 enzyme is expressed as early in the incipient primordia as GA2ox4. Because low GA levels are associated with maintaining the undifferentiated state of meristems, the activation of *GA2ox4* in these primordia may contribute more generally to the establishment of meristem identity in floral meristems. Indeed, the expression of *GA2ox4* precedes the establishment of meristematic identity as determined by *STM* and *WUS* expression, which are detectable in stage 2 floral primordia onwards (*Heisler et al., 2005*; *Long and Barton, 2000*; *Mayer et al., 1998*). Therefore, although STM is implicated in transcriptional activation of *GA2ox4* in the SAM (*Jasinski et al., 2005*), it is probably not responsible for its initial activation in early floral primordia. Instead, *GA2ox4* activation at this stage involves the MADS-box protein SVP, which contributes to transcriptional complexes that confer floral fate (*Balanzà et al., 2014*; *Gregis et al., 2009*; *Liu et al., 2009*).

## Perspectives

We describe at unprecedented resolution the changes in shape and cellular composition that occur at the shoot meristem during floral transition induced by the environmental cue of day length. Our analysis is based on developing methods for image analysis of the epidermis at 2.5D using fixed and cleared meristematic samples. Further methodological development to enable imaging of deeper cell layers of the transition meristem, as has been performed on the inflorescence meristem (*Gaillochet et al., 2017*; *Ma et al., 2019*; *Serrano-Mislata et al., 2017*), and analysis at higher temporal resolution using live imaging will help define which cellular changes are the primary cause of meristem doming. Our experiments also demonstrate dynamic changes in temporal and spatial expression of GA metabolism enzymes controlled by the GRN that regulates flowering. Similar approaches can be used to analyze the roles of other phytohormones such as auxin and cytokinin and to image the machinery that controls the behavior of the stem cells. Such approaches will provide a more complete picture of how the environment influences meristem shape, size, and function in a developmentally programmed and stereotypical way during floral induction. Furthermore, although the regulatory machinery that regulates flowering responses to environment has diverged during evolution, the FT/FD pathway and meristematic doming are highly conserved (*Andrés and Coupland, 2012*; *Kwiatkowska, 2008*; *Tal et al., 2017*), suggesting that the processes described in *A. thaliana* might be relevant for a wide range of species.

## Materials and methods

**Key resources table**

| Reagent type (species) or resource | Designation | Source or reference | Identifiers | Additional information |
|---|---|---|---|---|
| Genetic reagent (*Arabidopsis thaliana*) | ga2ox1–1 (Col-0) | *Rieu et al., 2008a* | | |
| Genetic reagent (*Arabidopsis thaliana*) | ga2ox2–1 (Col-0) | *Rieu et al., 2008a* | | |
| Genetic reagent (*Arabidopsis thaliana*) | ga2ox3–1 (Col-0) | *Rieu et al., 2008a* | | |
| Genetic reagent (*Arabidopsis thaliana*) | ga2ox4–1 (Col-0) | *Rieu et al., 2008a* | | |

*Continued on next page*

*Continued*

| Reagent type (species) or resource | Designation | Source or reference | Identifiers | Additional information |
|---|---|---|---|---|
| Genetic reagent (*Arabidopsis thaliana*) | ga2ox4–3 (Col-0) | *Rieu et al., 2008a* | | |
| Genetic reagent (*Arabidopsis thaliana*) | ga2ox6–2 (Col-0) | *Rieu et al., 2008a* | | |
| Genetic reagent (*Arabidopsis thaliana*) | ga2ox7–2 (Col-0) | *Magome et al., 2008* | | |
| Genetic reagent (*Arabidopsis thaliana*) | ga2ox8 (WiscDsLox263B11) (Col-0) | *Mateos et al., 2015* | | |
| Genetic reagent (*Arabidopsis thaliana*) | ga20ox2–1 (Col-0) | *Rieu et al., 2008b* | | |
| Genetic reagent (*Arabidopsis thaliana*) | ga3ox3–1 (Col-0) | *Mitchum et al., 2006* | | |
| Genetic reagent (*Arabidopsis thaliana*) | soc1-2 (Col-0) | *Lee et al., 2000* | | |
| Genetic reagent (*Arabidopsis thaliana*) | svp-41 (Col-0) | *Hartmann et al., 2000* | | |
| Genetic reagent (*Arabidopsis thaliana*) | ft-10 tsf-1 (Col-0) | *Jang et al., 2009* | | |
| Genetic reagent (*Arabidopsis thaliana*) | KNAT1::GA2ox7 (Col-0) | *Porri et al., 2012* | | |
| Genetic reagent (*Arabidopsis thaliana*) | AP1::AP1-GFP (Col-0) | *Urbanus et al., 2009* | | |
| Genetic reagent (*Arabidopsis thaliana*) | SVP::SVP-GFP (Col-0) | *Gregis et al., 2013* | | |
| Genetic reagent (*Arabidopsis thaliana*) | 35S::SOC1:GR soc1-1 (Ler-0) | *Hyun et al., 2016* | | |
| Genetic reagent (*Arabidopsis thaliana*) | GA20ox2::VENUS-GA20ox2 (Col-0) | This study | | See Materials and methods, section 'Plasmid construction and plant transformation' |
| Genetic reagent (*Arabidopsis thaliana*) | GA2ox2::VENUS-GA2ox2 (Col-0) | This study | | See Materials and methods, section 'Plasmid construction and plant transformation' |
| Genetic reagent (*Arabidopsis thaliana*) | GA2ox4::VENUS-GA2ox4 (Col-0) | This study | | See Materials and methods, section 'Plasmid construction and plant transformation' |

*Continued on next page*

*Continued*

| Reagent type (species) or resource | Designation | Source or reference | Identifiers | Additional information |
|---|---|---|---|---|
| Genetic reagent (*Arabidopsis thaliana*) | *GA2ox6::VENUS-GA2ox6 (Col-0)* | This study | | See Materials and methods, section 'Plasmid construction and plant transformation' |
| Sequence-based reagent | Various oligonucleotides | This paper | Primers for cloning | See *Supplementary file 1* |
| Sequence-based reagent | Various oligonucleotides | This paper | Primers for RT-qPCR | See *Supplementary file 2* |
| Sequence-based reagent | Various oligonucleotides | This paper | Primers for ChIP-qPCR | See *Supplementary file 4* |
| Chemical compound, drug | Renaissance 2200 | *Musielak et al., 2015* | | |
| Software, algorithm | | RStudio Team, 2015 | RRID:SCR_000432 | |
| Software, algorithm | MorphoGraphX | https://morphographx.org/ | | |
| Other | Fiji | doi:10.1038/nmeth.2019 | RRID:SCR_002285 | |

## Plant materials and growth condition

All plants used in this study were *Arabidopsis thaliana* Columbia (Col-0) background except for *35S:: SOC1:GR soc1-1* (*Hyun et al., 2016*), which is in Landsberg *erecta* (L*er*-0) genetic background. Mutant alleles were previously described: *ga2ox1*–1, *ga2ox2*–1, *ga2ox3*–1, *ga2ox4*–1, *ga2ox4*–3, *ga2ox6*–2 (*Rieu et al., 2008a*), *ga2ox7*–2 (*Magome et al., 2008*), *ga2ox8* (*WiscDsLox263B11*) (*Mateos et al., 2015*), *ga20ox2*–1 (*Rieu et al., 2008b*), *ga3ox3*–1 (*Mitchum et al., 2006*), *soc1-2* (*Lee et al., 2000*), *svp-41* (*Hartmann et al., 2000*), and *ft-10 tsf-1* (*Jang et al., 2009*). The following transgenic lines were used: *KNAT1::GA2ox7* (*Porri et al., 2012*), *AP1::AP1-GFP* (*Urbanus et al., 2009*), and *SVP::SVP-GFP* (*Gregis et al., 2013*). Plants were grown on soil under controlled conditions of SDs (8 hr light/16 hr dark) and LDs (16 hr light/8 hr dark). For GA and PAC treatment, plants were grown on growth medium containing Murashige and Skoog basal salts, 1% (w/v) sucrose, 0.05% (w/v) MES (pH 5.7), and 1% (w/v) agar.

## GA and PAC treatment

The $GA_3$ stock was prepared in 100% ethanol with a final concentration of 1 mM. The PAC stock was prepared in 100% dimethyl sulfoxide (DMSO) with a final concentration of 1 mM. GA and PAC treatments were performed on plants grown on MS medium. $GA_3$ or PAC solution (100 µM or 10 µM, respectively, dissolved in water) was applied directly to the shoot apices of these plants once 24 hr prior to harvesting.

## Dex treatment

Dex treatment was performed as described previously in *Hyun et al., 2016*.

## Plasmid construction and plant transformation

Full-length *GA20ox2*, *GA2ox2*, *GA2ox4*, and *GA2ox6* were amplified by PCR and cloned into the entry vector by BP or TOPO reaction (Invitrogen). Subsequently, 9× Ala Venus was introduced into the N-terminus of the coding sequences by the Polymerase Incomplete Primer Extension (PIPE) cloning method (*Klock and Lesley, 2009*). The entry clones were subcloned via LR reaction into the binary vectors, pEarlyGate or pGWB401, and the plasmids were then introduced into Agrobacterium strain GV3101 (pMP90RK) to transform Col-0, *ga20ox2*–1 or *ga2ox4*–3 mutants by floral dipping

(*Clough and Bent, 1998*). The primers used for plasmid construction are listed in *Supplementary file 1*.

## RNA extraction and real-time quantitative-PCR

Total RNA was isolated from plant tissues using the RNeasy plant mini kit (Qiagen) and treated with DNA-free DNA removal kit (Ambion) to remove residual genomic DNA. First-strand cDNA was synthesized using a Superscript II/IV First Strand Synthesis System (Invitrogen). Transcript levels were quantified by quantitative PCR in a CFX384 Touch Real-Time PCR Detection system (Bio-Rad) with GoTaq qPCR Master Mix (Promega) using the *PEX4* as the housekeeping gene (AT5G25760) to which data was normalized. Three technical replicates were performed for every biological replicate. The mean of three biological replicates with standard deviation is plotted graphically and the sequences of primers used for expression analyses are listed in *Supplementary file 2*.

## Chromatin-immunoprecipitation (ChIP) assays

ChIP analysis was performed as described previously in *Hyun et al., 2016* using primers listed in *Supplementary file 4*. To determine the fold enrichment levels, ChIP-DNA was quantified on a Roche Light Cycler 480 instrument (Roche) with iQ SYBR Green Supermix (BioRad) and normalized against *ACT8* (AT1G49240). The mean of three biological replicates with standard deviation is shown in graphs.

## Microscopy and imaging

Shoot apices at different developmental stages were dissected under a stereo microscope and fixed with 4% paraformaldehyde (PFA). The fixed samples were washed twice for 1 min in phosphate-buffered saline (PBS) and cleared with ClearSee (*Kurihara et al., 2015*) for 3–10 days at room temperature. The cell wall was stained with Renaissance 2200 [0.1% (v/v) in ClearSee] (*Musielak et al., 2015*) for 1–2 days, if necessary. Confocal microscopy was performed either on a LSM780 laser-scanning confocal microscope (Zeiss) or a TSC SP8 confocal microscope (Leica) as described previously (*Andres et al., 2015*; *Prunet, 2017*). The excitation wavelength was 405 nm for Renaissance and 514 nm for VENUS. The image collection was performed at 410–503 nm for Renaissance and 517–569 nm for VENUS. The Z intervals of sections in stacks were 2–3 µm for the maximum intensity projection or the optical sections and 0.5–1.0 µm for the surface analysis. To image GFP and VENUS fluorescence together, the lambda-mode images were obtained on a LSM780 using an excitation wavelength of 488 nm and emission wavelength of 495–635 nm. Spectral unmixing and processing of the obtained images were then conducted by ZEN imaging software (Zeiss) using GFP, VENUS, and the autofluorescence spectra as references. The autofluorescence spectrum was obtained by imaging a Col-0 meristem grown for 2 weeks in non-inductive SDs and transferred to inductive LDs for 5 days. Image analysis was performed by FIJI (https://fiji.sc/), to obtain maximum intensity projection images and optical sections of the confocal image stacks. Brightness and contrast were adjusted when necessary and to the same extent in the GFP and the VENUS channels. At least three experiments were conducted where more than three meristems of three individual transformants were observed.

## Image processing and analysis

The Z stacks of SAMs were acquired with a step size of 0.4 µm and converted to. tif files with FIJI. Using the MorphoGraphX (MGX) software (https://morphographx.org/) (*Barbier de Reuille et al., 2015*; *Kierzkowski et al., 2012*) the surface of the meristem was extracted and the Renaissance signal of the cell wall from the outer cell layer (L1) was projected and used for segmentation of the images. Cells were auto-segmented and corrected manually. The geometry of the surface was displayed as Gaussian curvatures with a neighboring of 10 µm. The boundary between the meristem and the developing primordia ($P_n$) was defined by negative Gaussian curvatures. The central cell of the meristem was selected from the summit of the meristem. The CZ and the adjacent PZ were defined by the location of the cells: zero to two cells from the central cell and three to five cells from the central cell, respectively. The curvilinear distance between primordia was measured using the Bezier process. The signal intensity maps were generated by extracting the fluorescent signal at the

surface of the meristem (2–10 µm from the meristem surface) and by projecting it onto the cellular mesh.

### Statistical analysis

Data were analyzed using ANOVA followed by a post-hoc Tukey's test using R software (http://r-project.org/).

## Acknowledgements

We thank John Chandler and Martina Cerise for comments on the manuscript. We thank the Central Microscopy Unit and IT Department of the Max Planck Institute for Plant Breeding Research for support.

## Additional information

### Funding

| Funder | Grant reference number | Author |
|---|---|---|
| Alexander von Humboldt-Stiftung | | Atsuko Kinoshita |
| Japanese Society for the Promotion of Science | | Atsuko Kinoshita |
| Deutsche Forschungsgemeinschaft | 390686111 | George Coupland |
| Max-Planck-Gesellschaft | Open-access funding | George Coupland |

The funders had no role in study design, data collection and interpretation, or the decision to submit the work for publication.

### Author contributions

Atsuko Kinoshita, Conceptualization, Formal analysis, Funding acquisition, Investigation, Methodology, Writing - original draft, Writing - review and editing, Acquisition of data; Alice Vayssières, René Richter, Qing Sang, Adrian Roggen, Annabel D van Driel, Formal analysis, Acquisition of data; Richard S Smith, Conceptualization; George Coupland, Conceptualization, Supervision, Funding acquisition, Investigation, Writing - original draft, Project administration, Writing - review and editing

### Author ORCIDs

Atsuko Kinoshita ⓘ https://orcid.org/0000-0001-9095-389X
René Richter ⓘ https://orcid.org/0000-0001-9833-2211
Annabel D van Driel ⓘ http://orcid.org/0000-0002-1629-5961
Richard S Smith ⓘ http://orcid.org/0000-0001-9220-0787
George Coupland ⓘ https://orcid.org/0000-0001-6988-4172

### Decision letter and Author response

Decision letter https://doi.org/10.7554/eLife.60661.sa1
Author response https://doi.org/10.7554/eLife.60661.sa2

## Additional files

### Supplementary files

- Supplementary file 1. Primers used for cloning in this study.
- Supplementary file 2. Primers used for RT-qPCR in this study.
- Supplementary file 3. Putative CArG-boxes and positions in *GA2ox4*.
- Supplementary file 4. Primers used for ChIP-qPCR for *GA2ox4*.

• Transparent reporting form

### Data availability

All data generated this study are included in the manuscript and supporting files. Source data files have been provided for Figures 1, 3, 6 and 7 and 8.

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
