## [Decision Letter]

**Acceptance summary:**

In this manuscript, you report a comprehensive analysis of morphological changes in the shoot apical meristem (SAM) during the floral transition, and analyze their relationship with the phytohormone GA pathway and two well-known flowering time genes in Arabidopsis. This study reveals important links among dynamic cellular changes, the flowering regulatory network and GA signaling in the SAM, which provide new insights into understanding the underlying mechanisms of the floral transition.

**Decision letter after peer review:**

Thank you for submitting your article " Regulation of shoot meristem shape by photoperiodic signaling and phytohormones during floral induction of Arabidopsis" for consideration by *eLife*. Your article has been reviewed by three peer reviewers, and the evaluation has been overseen by Hao Yu as the Reviewing Editor and Christian Hardtke as the Senior Editor.

The reviewers have discussed the reviews with one another and the Reviewing Editor has drafted this decision to help you prepare a revised submission.

Summary:

In this manuscript, the authors report a comprehensive analysis of morphological changes in the shoot apical meristem (SAM) during the floral transition, the associated dynamic modulation of expression of the enzymes involved in GA biosynthesis and catabolism, and their relationship with two flowering time genes, SOC1 and SVP, in Arabidopsis. This study reveals interesting links among dynamic cellular changes, the flowering regulatory network and GA signaling in the SAM during the floral transition.

Essential revisions:

While the topic and the findings are very relevant and of considerable interest, some concerns on the methodologies, image quality and interpretation of results as listed below need to be addressed to support the conclusions.

1) Studies on morphological and cellular features of the SAM during the floral transition should be better integrated with the analysis of molecular and genetic interactions between SOC1/SVP and GA metabolism genes. This will strengthen the novelty of this study through providing a clearer link among cellular/hormonal changes, the flowering regulatory network, and the associated morphological feature-SAM doming during the floral transition.

2) The quality of images in several figures, such as Figures 4, 6, Figure 2—figure supplement 3, and Figure 4—figure supplement 5, needs to be improved. The authors may consider replacing them with better images and/or providing relevant quantitative analysis.

3) Interpretation of some figures, such as Figures 3, 5 and Figure 4—figure supplement 6, needs to be revised to describe the results herein more accurately. For key images like Figure 5, presentation of images from biological replicates will be helpful.

Please also take into consideration the other specific comments from the reviewers below to revise the manuscript.

Reviewer #1:

The floral transition is a crucial process in plant development. During this transition, the shoot apical meristem stops producing leaves and starts making floral buds. The gene regulatory network (GRN) that controls the floral transition has been extensively studied, but much less is known about the cellular changes that underlie the transition from a vegetative to a reproductive meristem, and how they are linked to the flowering GRN. The manuscript by Kinoshita et al. reports a comprehensive analysis of morphological changes of the shoot meristem during the floral transition, the role of hormonal signaling in these changes and how the hormonal changes are linked to regulatory genes that control the floral transition.

Their overall conclusion is that the shoot meristem appears to go through two rapid transitions during the floral transition. First, there is a sudden increase in meristem size, associated with higher GA levels and signaling. Shortly afterwards, GA levels are reduced, coinciding with a change in the identity of organ primordia, from leaf primordia to floral buds; the regulatory genes SOC1 and SVP play a role in repressing GA signaling during the second transition. The work is thorough, the images are of very high quality and I consider the results of general interest.

My main criticisms are:

1) Currently, the manuscript seems made of two parts that are insufficiently connected. The first part is a detailed analysis of the early changes in meristem structure during the floral transition, focusing on cellular structure and on the role of GA metabolism and signaling. The processes studied in the first part were formally placed downstream of the floral transition by showing that they were triggered by a shift from short to long days and required the function of the FT/TSF genes. The second part is an analysis of the molecular and genetic interactions between SOC1/SVP and GA metabolism genes. In the Abstract and in Figure 9—figure supplement 2, both parts are linked. However, the experiments on SOC1 and SVP focus on the transition from leaf primordia to floral buds, assessed mostly as changes in the number of cauline leaves. It would have been more logical to test the effect of SVP/SOC directly on the processes studied in part 1, such as meristem size and number of cells in the central and peripheral zones during the floral transition. Similarly, it would have been better to show the effect of SOC1/SVP on the expression of GA20ox2 or GA20x4 using the same conditions as in Figures 1-6 (SD followed by 3, 5 and 7 LD), rather than the different conditions used in Figures 7, 9 and Figure 6—figure supplement 1 (9, 11, 13, 15 and 17 LD). A better connection between the flowering GRN and processes that occur during meristem doming is important because to a large extent the novelty of the paper is in the link between cellular/hormonal changes and the flowering GRN.

2) In Figure 5, the authors show reduced RGA-GFP expression in the periphery of the shoot apex after the floral induction. The text refers to reduced expression in the peripheral zone, but the images appear to show a reduction mostly in organ primordia, which are not the same as the meristem PZ. Increased GA in organ primordia or in the PZ would have different implications. Still in Figure 5, panels E-H and I-L show different sections of the same four apices. This is somewhat repetitive; perhaps the space in the figure could be better used to show the reproducibility of changes in RGA-GFP expression across biological replicates – the color bars show that quantitative data are available, so the authors could show a statistical analysis of RGA-GP intensity in specific regions across all four biological replicates.

Reviewer #2:

A number of classical studies have analyzed the morphological changes of the SAM during floral transition. However, a detailed and quantitative cellular analysis was still lacking. This work exploits recent advances in bioimaging to fill this gap. The main conclusion is that two well-known flowering-time transcription factors directly control the expression of GA metabolism in specific regions of the shoot apex and that GA signaling is necessary to promote the increase in cell number and cell size that cause doming of the SAM. This conclusion, with the detailed analysis that supports it, represents a significant advance in the field. The major concerns that I would raise refer to the quality of some images, and to the interpretation of some pieces of evidence, all of which could be solved by selecting additional images and/or reducing the intensity of the claims.

1) It is a pity that the imaging is mostly focused on the epidermal L1 layer, not allowing to draw conclusions about the behavior of the RZ, for instance. Examples of the level of detail required in this type of analysis are found in recent articles: Ma et al., 2019; Gaillochet et al., 2017; Serrano-Mislata et al., 2017. And also in Figure 2 in this manuscript. The other figures that require considerable improvement are:

– Figure 5: the conclusion about the complementarity between the expression domains of RGA-GFP and 20ox-Venus or 2ox-Venus (ie "DELLA is downregulated in the PZ by a local increase in GA levels during flowering") is not completely supported by the photos shown. In fact, the increase in 20ox-venus is hardly seen here -contrary to what the authors show in previous images.

– Figure 4: it cannot be concluded that cells are more elongated in panel K than G. Better images and/or quantitative analysis must be provided for this.

– Panels in Figure 4—figure supplement 5 need to be improved. Close-up and orthogonal views must help to accurately define the spatial domain and intensity of GA2ox4 reporter activity.

– Figure 6: most images require to be improved. It is hard to differentiate shoot, leaves, and floral organs. VENUS-GA20ox2 activity is not observed in the rib zone of shoot apices induced to flower in contrast to Figure 2

– Figure 2—figure supplement 3: close-up images and/or orthogonal views must be provided to support that the VENUS-GA20ox2 signal is detected in meristem tissues at 2wSD3LD and especially at 3wSD3LD.

2) Some statements need modification or stronger proof:

– According to Figure 3 E-F, *ga20ox2*-1 cells are smaller in both CZ and PZ. Thus, the text must be corrected and the conclusion that GA20ox2 activity specifically contributes to cell size increase in the PZ must be revisited. I suggest to remove this separation of cells in the CZ and PZ. An alternative could be to directly analyze cell sizes in VENUS-GA20ox2 apices. In this way, cell and non-cell autonomous effects of GA20ox2 activity could be assessed.

– According to Figure 4—figure supplement 6 panels D-F, cells in the CZ of ga2ox4-3 meristems are bigger with respect to the wt after 5 LDs. However, cell sizes are not affected in the surrounding PZ. This is difficult to understand since GA2ox4 activity is excluded from both domains at this stage. Again, removing the analysis of separated SAM cell populations would simplify the analysis without losing relevant information.

– KNAT1 is expressed in the SAM central region and rib zone but not in the epidermis (Lincoln et al., 1994; Porri et al., 2012). This should be considered for the interpretation of the results of this work. For example, mechanical constraints from inner tissues may restrict cell growth and division in the epidermis of KNAT1p:GA2ox7 apices. On the other hand, it is surprising that the difference in meristem width between KNAT1p:GA2ox7 and *ft tsf* apices is not reflected in the distance between primordia (Figure 1—figure supplement 1C).

Reviewer #3:

The manuscript by Kinoshita et al. entitled "Regulation of shoot meristem shape by photoperiodic signaling and phytohormones during floral induction of Arabidopsis" reports an cellular resolution analyses of the shoot apical meristem (SAM) shape change during floral transition. In particular, the authors found both cell number and size are transiently increased in response to inductive light conditions. Gibberellin (GA) has long been known to associated with floral induction. The authors analyzed the expression pattern of GA biosynthetic gene GA20ox2, GA metabolic gene GA2ox1 as well as a few related ones, and DELLA protein RGA, whose degradation is promoted by GA. The authors also characterized SAM cellular-level phenotypes of KNAT1::GA2ox7, *ga20ox2*-*1*, and several later flowering mutants. These analyses together suggest that GA promotes cell division as well as expansion, and GA signaling is mostly activated in the peripheral zone of the SAM. Another piece of new finding is that SOC1 and SVP both activates the expression GA2ox4 by direct binding to a promoter distal region. Taken together, this is an information rich manuscript that studied several aspects of floral induction with a focus on cellular level changes, especially its GA regulation. My suggestions are listed below.

1) This manuscript is more or less descriptive, although it describes cellular level changes. Ideally, it will be more informative if the authors can link the cellular level observations to SAM shape changes. Can the observed cell divisions and expansions explain SAM doming during floral transition? If not, what other changes are needed? I know this might be too much beyond the scope of the current research, but such a connection will really connect genetic information to geometry changes.

2) Both GA20ox2 and GA2ox1 have strong expression in abaxial side of leaf primordia. However, RGA is not present there. Can the authors comment on this?

3) It is unclear what “a”, “b” and “c” mean in several figures, such as Figure 1B-G.

---

## [Author Response]

Essential revisions:

While the topic and the findings are very relevant and of considerable interest, some concerns on the methodologies, image quality and interpretation of results as listed below need to be addressed to support the conclusions.

1) Studies on morphological and cellular features of the SAM during the floral transition should be better integrated with the analysis of molecular and genetic interactions between SOC1/SVP and GA metabolism genes. This will strengthen the novelty of this study through providing a clearer link among cellular/hormonal changes, the flowering regulatory network, and the associated morphological feature-SAM doming during the floral transition.

We incorporated new data and text to answer this point. In particular, we included imaging and segmentation analysis of the shoot meristem of *soc1-2* mutants after transfer from SD to LD, and compared the results with the analysis of Col, *ft tsf* mutants and *ga20ox2* mutants. These new data appear as Figure 7 and are described below under reviewer 1.

2) The quality of images in several figures, such as Figures 4, 6, Figure 2—figure supplement 3, and Figure 4—figure supplement 5, needs to be improved. The authors may consider replacing them with better images and/or providing relevant quantitative analysis.

We have improved these figures. The images in Figure 4 were improved. We replaced Figure 6 (now Figure 5) with entirely new images that are clearer. We edited Figure 2—figure supplement 3 to increase the size of the images and add a longitudinal view of each meristem. We edited Figure 4—figure supplement 5 to increase the size of the images and added a longitudinal view of each meristem.

3) Interpretation of some figures, such as Figures 3, 5 and Figure 4—figure supplement 6, needs to be revised to describe the results herein more accurately. For key images like Figure 5, presentation of images from biological replicates will be helpful.

We edited the text referring to Figure 3, as described in detail in the response to reviewer 2. We deleted Figure 5, because the analysis of RGA protein represented a small part of the manuscript that was not essential to our argument, while the resolution of the images was low and we could not substantially improve them in the time available. We discuss this further in the responses to reviewer 1 and reviewer 2. We comment on the interpretation of Figure 4—figure supplement 6 in the response to reviewer 2. We have now provided replicates for many figures.

Please also take into consideration the other specific comments from the reviewers below to revise the manuscript.

Reviewer #1:

The floral transition is a crucial process in plant development. During this transition, the shoot apical meristem stops producing leaves and starts making floral buds. The gene regulatory network (GRN) that controls the floral transition has been extensively studied, but much less is known about the cellular changes that underlie the transition from a vegetative to a reproductive meristem, and how they are linked to the flowering GRN. The manuscript by Kinoshita et al. reports a comprehensive analysis of morphological changes of the shoot meristem during the floral transition, the role of hormonal signaling in these changes and how the hormonal changes are linked to regulatory genes that control the floral transition.

Their overall conclusion is that the shoot meristem appears to go through two rapid transitions during the floral transition. First, there is a sudden increase in meristem size, associated with higher GA levels and signaling. Shortly afterwards, GA levels are reduced, coinciding with a change in the identity of organ primordia, from leaf primordia to floral buds; the regulatory genes SOC1 and SVP play a role in repressing GA signaling during the second transition. The work is thorough, the images are of very high quality and I consider the results of general interest.

My main criticisms are:

1) Currently, the manuscript seems made of two parts that are insufficiently connected. The first part is a detailed analysis of the early changes in meristem structure during the floral transition, focusing on cellular structure and on the role of GA metabolism and signaling. The processes studied in the first part were formally placed downstream of the floral transition by showing that they were triggered by a shift from short to long days and required the function of the FT/TSF genes. The second part is an analysis of the molecular and genetic interactions between SOC1/SVP and GA metabolism genes. In the Abstract and in Figure 9—figure supplement 2, both parts are linked. However, the experiments on SOC1 and SVP focus on the transition from leaf primordia to floral buds, assessed mostly as changes in the number of cauline leaves. It would have been more logical to test the effect of SVP/SOC directly on the processes studied in part 1, such as meristem size and number of cells in the central and peripheral zones during the floral transition. Similarly, it would have been better to show the effect of SOC1/SVP on the expression of GA20ox2 or GA20x4 using the same conditions as in Figures 1-6 (SD followed by 3, 5 and 7 LD), rather than the different conditions used in Figures 7, 9 and Figure 6—figure supplement 1 (9, 11, 13, 15 and 17 LD). A better connection between the flowering GRN and processes that occur during meristem doming is important because to a large extent the novelty of the paper is in the link between cellular/hormonal changes and the flowering GRN.

The reviewer finds that the manuscript is structured in two parts that are not sufficiently well integrated. The first part is the description of the structural changes of the shoot meristem during floral transition, and the second is the link between the GRN, particularly SOC1 and SVP, and the GA metabolism genes. The reviewer requests that these two parts are better integrated and concludes “A better connection between the flowering GRN and processes that occur during meristem doming is important because to a large extent the novelty of the paper is in the link between cellular/hormonal changes and the flowering GRN.” The reviewer makes suggestions for how this could be achieved, including “to test the effect of SVP/SOC directly on the processes studied in part 1, such as meristem size and number of cells in the central and peripheral zones during the floral transition”. We have now analyzed the meristem of *soc1-2* mutants after growth for 2 weeks under SDs and transfer to LDs, and compared these with wild-type plants, as done already in the original submission for *ft tsf* mutants. These data are presented in new Figure 7, described in the Results and Discussion. We could not carry out these experiments for *svp* mutants, because the plants flower extremely early under SDs, and the SAM is already doming by 2 weeks under SDs. Nevertheless, we believe that the inclusion of the *soc1* mutant data helps to integrate the different themes of the manuscript, as requested by the reviewer. Particularly, as these data appear in Figure 7, late in the Results section, and are compared with the results for *ft tsf* and *ga20ox2* presented earlier in the manuscript.

2) In Figure 5, the authors show reduced RGA-GFP expression in the periphery of the shoot apex after the floral induction. The text refers to reduced expression in the peripheral zone, but the images appear to show a reduction mostly in organ primordia, which are not the same as the meristem PZ. Increased GA in organ primordia or in the PZ would have different implications. Still in Figure 5, panels E-H and I-L show different sections of the same four apices. This is somewhat repetitive; perhaps the space in the figure could be better used to show the reproducibility of changes in RGA-GFP expression across biological replicates – the color bars show that quantitative data are available, so the authors could show a statistical analysis of RGA-GP intensity in specific regions across all four biological replicates.

The reviewer comments on the analysis of RGA-GFP shown in Figure 5, and raises several issues on the quality of these images and the relative quantification of RGAGFP and VENUS-GA2ox4 or VENUS-GA20ox2. The analysis of RGA-GFP was a relatively small part of the manuscript, and we could not rigorously quantify these low resolution images nor substantially improve them with the replicates. Furthermore, we could not repeat the experiment within a reasonable revision period. Therefore, as the analysis of RGA-GFP was a relatively small part of the manuscript and did not represent one of our major conclusions, we decided to delete Figure 5 and the description of the RGA-GFP data from the Results.

Reviewer #2:

A number of classical studies have analyzed the morphological changes of the SAM during floral transition. However, a detailed and quantitative cellular analysis was still lacking. This work exploits recent advances in bioimaging to fill this gap. The main conclusion is that two well-known flowering-time transcription factors directly control the expression of GA metabolism in specific regions of the shoot apex and that GA signaling is necessary to promote the increase in cell number and cell size that cause doming of the SAM. This conclusion, with the detailed analysis that supports it, represents a significant advance in the field. The major concerns that I would raise refer to the quality of some images, and to the interpretation of some pieces of evidence, all of which could be solved by selecting additional images and/or reducing the intensity of the claims.

1) It is a pity that the imaging is mostly focused on the epidermal L1 layer, not allowing to draw conclusions about the behavior of the RZ, for instance. Examples of the level of detail required in this type of analysis are found in recent articles: Ma et al., 2019; Gaillochet et al., 2017; Serrano-Mislata et al., 2017. And also in Figure 2 in this manuscript.

We agree with the reviewer that in the future it will be interesting to extend the current analysis to deeper meristematic cell layers, and we did make this point in the Discussion. We had cited two of the references mentioned by the reviewer, and now cite all three. Nevertheless, we believe that the current detailed analysis of cell size and number during the transition considerably extends current knowledge of the doming process.

The other figures that require considerable improvement are:

– Figure 5: the conclusion about the complementarity between the expression domains of RGA-GFP and 20ox-Venus or 2ox-Venus (ie "DELLA is downregulated in the PZ by a local increase in GA levels during flowering") is not completely supported by the photos shown. In fact, the increase in 20ox-venus is hardly seen here -contrary to what the authors show in previous images.

This comment is similar to the one made by reviewer 1. As described above, we have now deleted Figure 5, as we could not improve these images during the revision and they contributed a small part of our overall argument.

– Figure 4: it cannot be concluded that cells are more elongated in panel K than G. Better images and/or quantitative analysis must be provided for this.

We have improved these images by increasing the brightness and contrast of the Renaissance channel thereby increasing the visibility of the cells. We did not change the GFP channel.

– Panels in Figure 4—figure supplement 5 need to be improved. Close-up and orthogonal views must help to accurately define the spatial domain and intensity of GA2ox4 reporter activity.

We have edited Figure 4—figure supplement 5 to include close up and longitudinal views.

– Figure 6: most images require to be improved. It is hard to differentiate shoot, leaves, and floral organs. VENUS-GA20ox2 activity is not observed in the rib zone of shoot apices induced to flower in contrast to Figure 2

We have replaced Figure 6 (now Figure 5) entirely with new images that show the relationship between AP1-GFP and VENUS-GA2ox4 in the vegetative, transition and inflorescence meristem. Apices are viewed from the top and from the side so that leaves and floral primordia are easily distinguished.

– Figure 2—figure supplement 3: close-up images and/or orthogonal views must be provided to support that the VENUS-GA20ox2 signal is detected in meristem tissues at 2wSD3LD and especially at 3wSD3LD.

We have edited Figure 2—figure supplement 3 to include close up and longitudinal views.

2) Some statements need modification or stronger proof:

– According to Figure 3 E-F, ga20ox2-1 cells are smaller in both CZ and PZ. Thus, the text must be corrected and the conclusion that GA20ox2 activity specifically contributes to cell size increase in the PZ must be revisited. I suggest to remove this separation of cells in the CZ and PZ. An alternative could be to directly analyze cell sizes in VENUS-GA20ox2 apices. In this way, cell and non-cell autonomous effects of GA20ox2 activity could be assessed.

We have edited this text as requested by the reviewer. We have made it clear in the text that smaller cells of *ga20ox-2* specifically in the PZ are detected at the 5LD time point during the floral transition, following on from the previous sentence that referred to meristem size at this time point. (“Analysis of different regions of the meristem, as described previously in Figure S2, at 5LD detected a reduced number of cells and the presence of smaller cells in the ring outside the central region (Figure 3B-F)”). We have also made it clear that at 9LDs cells are smaller both in the CZ and PZ (“although at 9LD cell size was larger in the CZ and PZ of *ga20ox2-1* meristems than in those of Col-0 plants (Figure 3E, F).”). We prefer to maintain the measurements in the CZ and PZ as differences are detected at specific time points, as now made clear in the edited text.

– According to Figure 4—figure supplement 6 panels D-F, cells in the CZ of ga2ox4-3 meristems are bigger with respect to the wt after 5 LDs. However, cell sizes are not affected in the surrounding PZ. This is difficult to understand since GA2ox4 activity is excluded from both domains at this stage. Again, removing the analysis of separated SAM cell populations would simplify the analysis without losing relevant information.

This seems to be a misunderstanding, because no significant difference in cell size was detected in the CZ of *ga2ox4-3* compared to Col-0 in Figure 4—figure supplement 6E.

– KNAT1 is expressed in the SAM central region and rib zone but not in the epidermis (Lincoln et al., 1994; Porri et al., 2012). This should be considered for the interpretation of the results of this work. For example, mechanical constraints from inner tissues may restrict cell growth and division in the epidermis of KNAT1p:GA2ox7 apices. On the other hand, it is surprising that the difference in meristem width between KNAT1p:GA2ox7 and ft tsf apices is not reflected in the distance between primordia (Figure 1—figure supplement 1C).

We have now mentioned that *KNAT1* is not expressed in the epidermis in the Results. *“*This transgene reduces the level of GA precursor specifically in the SAM, but is not expressed directly in the epidermis (Lincoln et al., 1994; Porri et al., 2012; Schomburg et al., 2003).” Also, in the Discussion we raise the specific point mentioned by the reviewer on mechanical constraints. (”The *KNAT1* promoter is not active in the epidermis (Lincoln et al., 1994; Porri et al., 2012), so reducing GA levels in the inner layers of the SAM must lead to the reductions in cell size and cell number detected in the epidermis, perhaps due to indirectly reducing GA levels in the epidermis or to mechanical constraints.”). Concerning the difference in distance between primordia and width in the apices of *KNAT1p:GA2ox7 and ft tsf,* we have now included in Figure 1—figure supplement 1 the width measurements (Figure 1—figure supplement 1D and E), as these were not shown in the submitted version. The width of *ft tsf* apices is significantly greater than *KNAT1p:GA2ox7* only at 7LDs, and at this time point there is no difference in distance between primordia. We assume that this is because the *apices of KNAT1p:GA2ox7* are domed at 7LDs. As the distance between primordia is measured across the top of the apex, the increase in height of *KNAT1p:GA2ox7* apices presumably compensates for the reduction in width.

Reviewer #3:

The manuscript by Kinoshita et al. entitled "Regulation of shoot meristem shape by photoperiodic signaling and phytohormones during floral induction of Arabidopsis" reports an cellular resolution analyses of the shoot apical meristem (SAM) shape change during floral transition. In particular, the authors found both cell number and size are transiently increased in response to inductive light conditions. Gibberellin (GA) has long been known to associated with floral induction. The authors analyzed the expression pattern of GA biosynthetic gene GA20ox2, GA metabolic gene GA2ox1 as well as a few related ones, and DELLA protein RGA, whose degradation is promoted by GA. The authors also characterized SAM cellular-level phenotypes of KNAT1::GA2ox7, ga20ox2-1, and several later flowering mutants. These analyses together suggest that GA promotes cell division as well as expansion, and GA signaling is mostly activated in the peripheral zone of the SAM. Another piece of new finding is that SOC1 and SVP both activates the expression GA2ox4 by direct binding to a promoter distal region. Taken together, this is an information rich manuscript that studied several aspects of floral induction with a focus on cellular level changes, especially its GA regulation. My suggestions are listed below.

1) This manuscript is more or less descriptive, although it describes cellular level changes. Ideally, it will be more informative if the authors can link the cellular level observations to SAM shape changes. Can the observed cell divisions and expansions explain SAM doming during floral transition? If not, what other changes are needed? I know this might be too much beyond the scope of the current research, but such a connection will really connect genetic information to geometry changes.

We agree with the reviewer that the long-term objective is to explain which cellular changes confer doming and how these are regulated. However, as conceded by the reviewer, this is “beyond the scope of the current research”, which begins the quantitative analysis of doming by defining cell numbers and sizes through the process, and linking these to the GRN controlling flowering and to GA metabolism. This issue is related to that of reviewer 2 on the importance of imaging deeper meristematic layers. In the edited Discussion where we address imaging of deeper layers, we also emphasise the future direction of determining the cellular changes that are causal on doming.

2) Both GA20ox2 and GA2ox1 have strong expression in abaxial side of leaf primordia. However, RGA is not present there. Can the authors comment on this?

We deleted the images on RGA-GFP in Figure 5 as described above in response to the other reviewers.

3) It is unclear what “a”, “b” and “c” mean in several figures, such as Figure 1B-G.

In the figure legends we have defined the letters indicate statistically significant differences. This includes “a”, “b” and “c” in Figure 1B-G.